# Stochastic portfolio optimization: A regret-based approach on volatility risk measures: An empirical evidence from The New York stock market

**AmirMohammad Larni-Fooeik**, **Seyed Jafar Sadjadi, Emran Mohammadi** *

School of Industrial Engineering, Iran University of Science and Technology, Tehran, Iran

\* E_Mohammadi@iust.ac.ir

**Data Availability Statement:** All relevant data are within the manuscript.

**Funding:** The author(s) received no specific funding for this work.

## Abstract

Portfolio optimization involves finding the ideal combination of securities and shares to reduce risk and increase profit in an investment. To assess the impact of risk in portfolio optimization, we utilize a significant volatility risk measure series. Behavioral finance biases play a critical role in portfolio optimization and the efficient allocation of stocks. Regret, within the realm of behavioral finance, is the feeling of remorse that causes hesitation in making significant decisions and avoiding actions that could lead to poor investment choices. This behavior often leads investors to hold onto losing investments for extended periods, refusing to acknowledge mistakes and accept losses. Ironically, by evading regret, investors may miss out on potential opportunities. in this paper, our purpose is to compare investment scenarios in the decision-making process and calculate the amount of regret obtained in each scenario. To accomplish this, we consider volatility risk metrics and utilize stochastic optimization to identify the most suitable scenario that not only maximizes yield in the investment portfolio and minimizes risk, but also minimizes resulting regret. To convert each multi-objective model into a single objective, we employ the augmented epsilon constraint (AEC) method to establish the Pareto efficiency frontier. As a means of validating the solution of this method, we analyze data spanning 20, 50, and 100 weeks from 150 selected stocks in the New York market based on fundamental analysis. The results show that the selection of the mad risk measure in the time horizon of 100 weeks with a regret rate of 0.104 is the most appropriate research scenario. this article recommended that investors diversify their portfolios by investing in a variety of assets. This can help reduce risk and increase overall returns and improve financial literacy among investors.

## 1. Introduction

The advancement of economic growth poses significant opportunities and threats for all societies and governments. To address this situation, efficient roles are played by capital markets [1,2]. In both Theoretical and practical situations, the investment industry has experienced

**Competing interests:** The authors have declared that no competing interests exist.

significant growth in the last few decades. As a result of this growth, the financial markets are efficiently developed. Each year, billions of dollars are invested in various sectors by individual investors, brokers, and fund managers [3,4]. Generally, portfolio optimization refers to selecting investments in a way that spreads risk. To maximize returns, portfolio optimization involves selecting the most suitable and optimal number of stocks among a variety of types Considering risk minimization [5]. To consider both risk and return objectives in portfolio optimization problems, Markowitz's classical concept has made a significant contribution to the design of most financial models used for the selection of portfolios in the financial marketplace. Based on this concept, the two factors of return and risk, this model asserts that investors seek to minimize the expected variance of their portfolios at a certain level of their desired return [6,7]. Nevertheless, it remains crucial to highlight the selection of stocks that aim to minimize missed opportunities or regrets, while simultaneously considering the trade-off between risk and return. In essence, a judicious equilibrium between risk, return, and regret has been established.

Taking into account uncertainty and risk is crucial in portfolio optimization. By using techniques like stochastic programming, investors can account for the inherent uncertainty in the inputs and make more realistic and dependable decisions. This approach enhances the portfolio's performance and helps to mitigate potential suboptimal outcomes [8–10]. By considering the stochastic approach in the inputs, investors can make better-informed decisions and manage their risk more effectively. To enhance the performance of an investment portfolio effectively, it is important to consider a multitude of investment scenarios, including short-term, medium-term, and long-term options [11]. A short-term investment provides a rapid return, but it is often accompanied by heightened volatility and an increased level of risk for investors. Alternatively, medium-term investments offer a balance of risk and return. A long-term investment carries a lower risk; however, it requires a greater degree of patience to produce a substantial return. To determine the appropriate investment horizon, investors need to consider all three scenarios. Therefore, achieving a desirable balance between return and risk requires considering investment scenarios [12]. For an optimal stock portfolio, it is essential to select the right stocks, which can be achieved to some extent through fundamental analysis. fundamental analysis involves the examination of market data, including historical prices and trading volume, to evaluate securities. This approach is founded on the belief that price movements and market trends can provide insight into the future direction of a security's value. The primary objective of fundamental analysis is to identify potential buying or selling opportunities by analyzing patterns and trends in market data [13].

Risk measures play a crucial role in portfolio optimization. Volatility risk measures, such as standard deviation or variance, help investors quantify and manage the potential downside risk of their investments or portfolios. By incorporating these measures into the optimization process, investors can construct portfolios that align with their risk preferences. Diversifying risk across different asset classes further helps to reduce the overall portfolio risk [14]. By including assets with different levels of volatility in a portfolio, investors can reduce the overall volatility of the portfolio and potentially increase the risk-adjusted returns [15]. In this research, we use well-known volatility risk measures. They include Semi-Variance (SV) [16], Mean Absolute Deviation (Mad) [17], and Semi Absolute Devastation (SAD) [18]. The need to compare volatility risk measures arises when evaluating their performance within the presented model, as well as examining how their returns and regret vary across different time horizons. This aspect has received comparatively less attention in previous research, making it an important area to explore.

More specifically, the main goal of this article is to develop a scenario-based program that expands the concept of stochastic optimization, as it was proposed by Xidonas et al [19], to the

multi-objective case. xidonas et al introduce the concept of "regret" to identify robust solutions to optimization problems. Regret is the deviation of an obtained solution from the optimum solution according to a specific scenario of parameters. In other words, it can be defined as the difference between the obtained gain and the gain that we could get if we knew in advance which scenario would surely occur. Regret is the deviation of an obtained solution from the optimum solution according to a specific scenario of parameters. In other words, it can be defined as the difference between the obtained gain and the gain that we could get if we knew in advance which scenario would surely occur. To optimize stock portfolios through stochastic optimization, the risk objective function holds significance. Hence, we incorporate key volatility risk measures such as SV, MAD, and SAD across different investment scenarios: short-term, medium-term, and long-term. This method enables us to select the most suitable risk scenario and measure that minimizes regret or missed opportunities.

The contributions of our study to respond to the research gaps found are summarized below:

- a model has been presented that considers investors' regret for both the return and risk of their investments. The model functions in two sections: firstly, it evaluates the extent of regret for each aspect of risk and return, and then it sums up these two evaluations in the second stage. The resulting value indicates the level of regret associated with the selected stocks, ranging between 0 and 1. A lower value implies a smaller missed opportunity.

- To thoroughly analyze the level of regret, the study accounts for varying investment horizons among investors. Specific investment time scenarios have been identified: 20 weeks for the short term, 50 weeks for the medium term, and 100 weeks for the long term. Emphasizing the investment perspective enables us to identify the optimal time horizon that minimizes regrets when selecting investments.

- Taking into account historical returns for each stock, when relevant data is accessible, stochastic planning can be employed to calculate the level of regret and determine the appropriate allocation of weights for each stock. This method allows us to quantitatively represent the potential impact of each investment week in a probabilistic fashion, aiding in decision-making.

- In evaluating the presented model, three well-defined risk measures, sv (semi-variance), sad (semi-absolute deviation), and mad (mean absolute deviation), have been utilized, each taking into account three investment horizons. The research incorporates nine investment scenarios, empowering investors to select the optimal scenario by choosing the appropriate investment time horizon and risk measure. This approach aims to enhance investment profitability while simultaneously reducing the regret associated with unselected stocks.

- To validate the introduced model, historical data from 150 carefully selected stocks listed on the New York Stock Exchange (NYSE) has been employed. These stocks were chosen based on fundamental analysis criteria, enhancing the reliability and confidence level of the obtained results.

The remainder of the paper is organized as follows: In Section 2, we review the history and applications of portfolio optimization models that use the regret approach. In particular, we classify and present key research articles and theoretical notes on portfolio optimization, with a focus on the regret portfolio optimization approach. In Section 3, we present the regret modeling framework, then explain the fundamental application in extracting data and we extend the portfolio optimization with the regret approach by introducing the relevant volatility risk measures that have a significant impact in a multi-objective portfolio optimization

context. In Section 4, we test the proposed model with an illustrative application on the securities of the New York stock market, we consider 3 scenarios for portfolio selection that include 20,50, and 100-week investments, on the other hand, survey the short-term, mid-term, and long-term investments. Finally, in Section 5, key findings and conclusions are given intuitively.

## 2. Literature review

Over the past few decades, a variety of research on regret approach application in portfolio optimization has been conducted, which can be summarized as follows:

Giove et al [20] described how prices of securities are treated as interval variables in a portfolio selection problem, where a regret function is used to formulate an optimal decision-making procedure for a portfolio selection problem. In an attempt to minimize expected regret in each portfolio, Li et al [21] developed a model that minimizes the distance between the portfolio's maximum return and the actual return by applying expected regret minimization. Nwogugu [22] introduced the concepts of reducing return regrets, acknowledging losses, and framing as a way to reduce return regret. By considering future returns and creating Minimax regret portfolios based on the modern economic concept of the efficient frontier, Xidonas et al [19] found representative points on the efficient frontier. The weighted sum problem of maximum regrets and its properties were discussed by Rivaz and Yaghoobi [23] explained multi-objective linear programming and interval objective function coefficients for their research. Baule et al [24] have explained regret affects portfolio weights differently and will differ from Markowitz's model depending on what distributional characteristics make investments less or more attractive. Ji et al [25] reformulated polyhedral and conic support sets as an alternative to the worst-case regret optimization problem. The Tsionas [26] showed robust solutions could be encountered to the minmax regret problem, and similar Monte Carlo simulators were developed without establishing any scenarios. Li and Wang [27] developed the minimax regret criterion that can be used to select robust multi-objective portfolios in the face of ellipsoidal uncertainty sets. Greotzner and Werner [28] provided a robust definition of regret by broadening the concept's scope from a single objective to a multi-objective set of phenomena. Ding and Uryasevy [29] explained a new risk measure for portfolio performance called the expected regret of drawdown, which is based on the expected regret of a drawdown above the threshold. Stoltz and Lugosi [30] designed sequential investment strategies to minimize internal regrets. Gregory et al [31] identified a robust counterpart and evaluated its cost of robustness as part of their investigation of optimal portfolio optimization. Kagrecha et al [32] presented a stochastic multi-armed bandit setting in which constrained regret minimization over a given time frame is studied to solve the problem of constrained regret minimization. Deng and Geng [33] propose a novel and flexible two-parameter fuzzy number that he proposes which can be used by investors to capture their attitude toward the market (whether they are optimistic, pessimistic, or neutral). Khan et al [34] investigated the relationship between terrorism and stock market returns and volatility, specifically focusing on the context of Pakistan's stock exchange. By examining this dynamic, the study aims to shed light on the impact of terrorism on financial markets and provide empirical evidence to inform policymakers and investors. The article utilizes comprehensive data from Pakistan's stock exchange, considering both the occurrence and intensity of terrorist attacks. Through rigorous statistical analysis and econometric modeling, the study explores the causal linkages between terrorism events and stock market performance indicators. The findings of this research will contribute to the understanding of how terrorism affects the financial sector in a specific geopolitical setting and may offer insights into risk management strategies in similar contexts worldwide.

To assess the congruity with previous studies, a comprehensive literature review table has been incorporated in this research article. This table presents a comparative analysis of relevant articles about the research domain. Included in the table are various aspects considered in the reviewed studies, such as solution technique, investment constraints, model types, and uncertainties. The details of the literature review can be found in Table 1.

Based on our extensive investigation and exploration in this domain. After studying the previous studies according to the literature review table, we found that less has been addressed to the optimization of the possible stock portfolio considering regret, and these models have always been single-period or multi-period. Time scenarios are not considered for the time horizon. Therefore, in this article, we address these research gaps. This research article aims to address this gap by examining the application of the regret approach through the AEC technique for solving methods. Specifically, the study focuses on investigating the impact of portfolio investment's output and input on the problem at hand.

## 3. Proposed model and development

In this section, we present a comprehensive overview of the widely recognized issue of volatility risk measures SV, MAD, and SAD. We delve into the fundamental analysis of this problem, taking into account the concept of regret. Additionally, we employ the AEC method as a means to effectively address and resolve these problems. The schematic summary of all steps in the proposed optimization model is shown in Fig 1.

Following the aforementioned procedures, this section will delve into the evaluation of risk measures formulated using the regret-based approach. Table 2 presents an overview of the sets, parameters, and decision variables utilized for each volatility risk measure and regret-based model.

### 3.1 SV risk measure

Investment portfolio risk measurement using a variance measure raises the question of why fines and rewards should both be considered risks. As a solution, Harry Markowitz introduced a risk measure called the SV risk measure. There are several types of downside risk metrics, including SV. Under this category, risk is defined as values lower than the expected return. Below are the constraints that are included in this model [16]:

$$Min\ Z_1 = \frac{1}{T}\sum_{t=1}^{T} y_t^2 \tag{1}$$

$$Max\ Z_2 = \sum_{I=1}^{N} x_i \bar{R}_I \tag{2}$$

$$\sum_{i=1}^{n} \mu_i x_i = \mu_p$$

$$y_t \geq \mu_p - \sum_{i=1}^{n} R_{it} x_i \qquad\qquad t = 1, \ldots, T \tag{3}$$

$$y_t \geq 0 \qquad\qquad t = 1, \ldots, T \tag{4}$$

Eq 1 represents the objective function of risk minimization. Based on Eq 2, the portfolio's

**Table 1. Review of the most relevant documents.**

| # | Authors | Year | Exact solution | Heuristic algorithm | Metaheuristic algorithm | Simulation | Single period | Multi-period | Cardinality | Boundary | Transaction | Turnover | Others | Numerical | Case Study | Hypothetical | Model type | Certainly | Robust | Fuzzy | Stochastic | Others | Reference |
|---|---|---|---|---|---|---|---|---|---|---|---|---|---|---|---|---|---|---|---|---|---|---|---|
| 1 | Kagrecha et al | 2023 | | | bandit | | | ✓ | | ✓ | | | ✓ | ✓ | | | NLP | | | | ✓ | | [32] |
| 2 | Ding and Uryasev | 2022 | ✓ | | | | | ✓ | | ✓ | | | | ✓ | | | | ✓ | | | | | [29] |
| 3 | Groetzner and Werner | 2022 | ✓ | | | | ✓ | | ✓ | ✓ | | | | ✓ | | | MOLP | | ✓ | | | | [28] |
| 4 | Benati and Conde | 2022 | ✓ | | | | ✓ | | ✓ | ✓ | ✓ | ✓ | ✓ | ✓ | | | LP | | ✓ | | | | [35] |
| 5 | Caçador et al. | 2022 | | | GA | | ✓ | | ✓ | ✓ | | | | | ✓ | | LP | | ✓ | | | | [36] |
| 6 | Filho and Silva Neiro | 2022 | | | | MC | ✓ | | | | | | ✓ | ✓ | | | LP and NLP | | ✓ | | ✓ | | [37] |
| 7 | Li et al. | 2021 | ✓ | | | | | ✓ | | | ✓ | | | ✓ | | | LP | | ✓ | | | | [38] |
| 8 | Gong et al. | 2021 | ✓ | | | | ✓ | | ✓ | ✓ | | | | | ✓ | | LP | | | ✓ | | | [39] |
| 9 | Chakrabarti | 2021 | ✓ | | | | ✓ | | | ✓ | ✓ | | | | ✓ | | LP | | ✓ | | | | [40] |
| 10 | Caçador et al. | 2021 | ✓ | | | | ✓ | | ✓ | ✓ | ✓ | | ✓ | | ✓ | | LP | | ✓ | | | | [41] |
| 11 | Won and Kim | 2020 | ✓ | | | | ✓ | | ✓ | ✓ | ✓ | ✓ | | ✓ | | | LP | | ✓ | | | | [42] |
| 12 | Li and Wang | 2020 | ✓ | | | | | ✓ | ✓ | ✓ | | | | ✓ | | | LP | | ✓ | | | | [27] |
| 13 | Hernandez and al Janabi | 2020 | ✓ | | | | ✓ | | | | | | | ✓ | | | LP and NLP | ✓ | | | | | [43] |
| 14 | Caçador et al | 2020 | ✓ | | | | ✓ | | ✓ | ✓ | | ✓ | | | ✓ | | LP | | ✓ | | | | [44] |
| 15 | Vohra and Fabozzi | 2019 | ✓ | | | | ✓ | | | ✓ | | | ✓ | | ✓ | | LP | ✓ | | | | | [45] |
| 16 | Baule et al. | 2019 | ✓ | | | ✓ | ✓ | | | | | | | ✓ | | | LP | ✓ | | | | | [24] |
| 17 | Huang et al. | 2018 | ✓ | | | | | ✓ | | | ✓ | | | | ✓ | | LP | ✓ | | | | | [46] |
| 18 | Van den Broeke et al. | 2018 | ✓ | | | | | | | ✓ | | ✓ | | | ✓ | | MILP | ✓ | ✓ | | | | [47] |
| 19 | Simões et al. | 2018 | ✓ | | | | ✓ | | ✓ | ✓ | | | | | ✓ | | LP | | ✓ | | | | [48] |
| 20 | Rivaz and Yaghoobi | 2018 | ✓ | | | | ✓ | | | ✓ | | | ✓ | ✓ | | | MOLP | ✓ | | | | | [49] |
| 21 | Xidonas, et al.(b) | 2017 | ✓ | | | | | ✓ | ✓ | ✓ | | | | | ✓ | | MINLP | | ✓ | | | | [19] |
| 22 | Xidonas, et al.(a) | 2017 | ✓ | | | | | ✓ | | | | | | | ✓ | | MILP | | ✓ | | | | [11] |
| 23 | Mohr and Dochow | 2017 | ✓ | | | | | ✓ | | ✓ | | | | | ✓ | | MILP | ✓ | | | | | [50] |
| 24 | Grechuk and Zabarankin | 2017 | | | GA | | ✓ | | | | | | ✓ | | | | LP | | ✓ | | | | [14] |
| 25 | Fernandez et al. | 2013 | | | | | | ✓ | ✓ | | | | | ✓ | | | NLP | | ✓ | ✓ | | | [51] |
| 26 | Lourenço et al. | 2012 | | | | | | ✓ | | ✓ | | | | | ✓ | | MILP | | ✓ | | | | [52] |
| 27 | Bean and Singer | 2012 | | ✓ | | ✓ | | ✓ | ✓ | | ✓ | | | | | ✓ | MILP | | | | | | [53] |
| 28 | Gregory et al. | 2011 | ✓ | | | | | ✓ | | ✓ | | | | ✓ | | | MILP | ✓ | | | | | [31] |
| 29 | Giove et al. | 2006 | ✓ | | | | ✓ | | | ✓ | | | | ✓ | | | LP | | | ✓ | | | [20] |
| 30 | Nwogugu | 2006 | ✓ | | | | ✓ | | | ✓ | | | | ✓ | | | NLP | ✓ | | | | | [22] |
| 31 | Larni-Fooeik et al | 2024 | ✓ | | | | | SBA | ✓ | ✓ | | | ✓ | | ✓ | | MILP | | | | ✓ | | |

Genetic Algorithm (GA), Mont Carlo (MC), Linear Programming (LP), Mult objective Linear Programming (MOLP), Non-Linear Programming (NLP), Mix Integer Linear Programming (MILP), Mix Integer Non-Linear Programming (MINLP), Scenario-Based Approach (SBA).

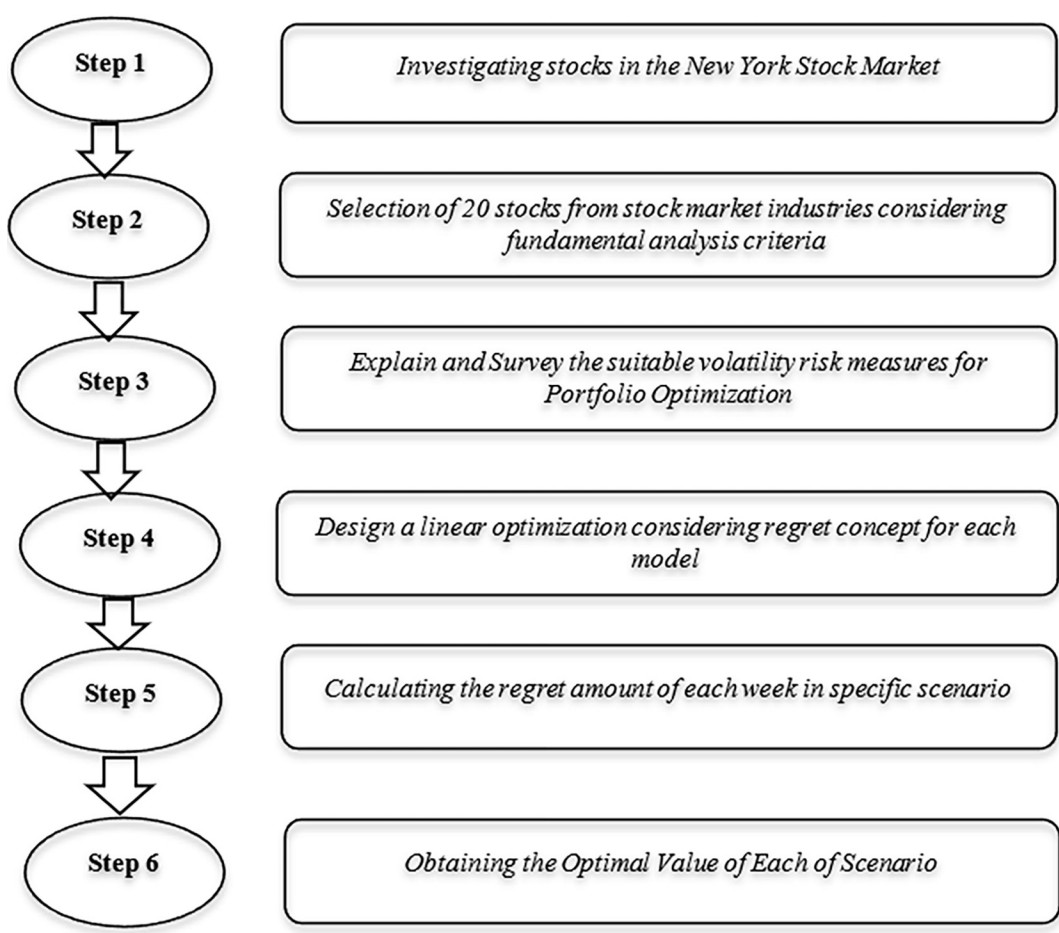

**Fig 1. The schematic summary of all steps in the proposed optimization model.**

expected return is calculated by averaging the selected stocks. In the objective function, Eqs 3 and 4 are used to solve the relationship between absolute value and function. For the objective function to be stripped of absolute values, the relation $y_t$ is written in a smaller form equal to and a larger form equal to $\mu_p - \sum_{i=1}^{n} r_{it} x_i$, but because values above the expected return do not count as risks, relation 4 is converted to the form $y_t \geq 0$.

## 3.2 MAD risk measure

The absolute deviation risk measure is a statistical metric used in portfolio optimization to quantify the risk associated with the return variability. It provides a measure of the dispersion of returns from the average or expected return, regardless of the direction of deviation. Unlike other risk measures such as variance and standard deviation, which focus on squared deviations, the absolute deviation risk measure considers deviations in their absolute form. Absolute deviation risk measures are used in portfolio optimization to assess downside risk or potential losses an investor might experience. Due to its ability to capture both positive and negative deviations from the mean, it suits investors who care more about losses than volatility overall [17]. Eqs 5 to 9 represent the MAD risk measures model. Eq 5 represents the objective function of risk minimization and Eq 6 represents the objective function of profit maximization. Eqs 8 and 9 were used for linearization of Eq 5. To transform the risk objective function, which

**Table 2. Sets, parameters, and decision variables of the proposed model.**

| sets | i | selected stocks from the New York Stock Exchange | i = 1, . . ., 20 |
|---|---|---|---|
| | j | Selected Investment weeks | j = 1, . . ., 20, . . . 50, . . ., 100 |
| | s | Scenario of stocks investing | s = 1, . . ., 20, . . . 50, . . ., 100 |
| | T | Historical selected week for each scenario | T = 1, . . ., 20, . . . 50, . . ., 100 |
| **parameters** | $Z1_s$ | The solution for the risk objective function in each scenario for risk objective function | |
| | $Z1_s^*$ | The optimal solution for the risk objective function | |
| | $Z2_s$ | The solution for the Return objective function in each scenario for the risk objective function | |
| | $Z2_s^*$ | The optimal solution for the Return objective function | |
| | $R_{it}$ | The return of the $i-th$ stock in the $t-th$ time | |
| | $\bar{R}_I^-$ | The average return of the $i-th$ stock for each scenario | |
| | $\mu_p$ | The weighted average return of the entire portfolio | |
| | $\mu_i$ | The weighted average return of the selected stocks | |
| **Decision variable** | $x_i$ | The amount of the $i-th$ stock selected in the portfolio | |
| | $z_1$ | Risk-related objective function | |
| | $z_2$ | Return-related objective function | |
| | $y_t$ | An additional positive variable in the risk objective function which is used for the linearization of an absolute value | |
| | $\pi_s$ | Probability of scenario for each week | |
| | $\eta1$ | Coefficient factor between the maximum regret of objective function risk and the value of objective function risk | |
| | $\eta2$ | Coefficient factor between the maximum regret of objective function Return and the value of objective function Return | |
| | REG1 | maximum relative regret of objective function of risk | $0 \leq REG1 \leq 1$ |
| | REG2 | maximum relative regret of objective function of Return | $0 \leq REG2 \leq 1$ |
| | REGT | Summation of maximum relative regret of objective function of risk and Return | $0 \leq REGT \leq 1$ |

includes an absolute value, into a linear form, a variable $y_t$ is utilized.

$$Min\ Z_1 = \frac{1}{T}\sum_{i=1}^{T}\left|\sum_{i=1}^{N}(R_{it} - \bar{R}_I)\right| = Min\ Z_1 \frac{1}{T}\sum_{i=1}^{T}y_t \tag{5}$$

$$Max\ Z_2 = \sum_{I=1}^{N}x_i\bar{R}_I \tag{6}$$

$$\sum_{i=1}^{n}\mu_i x_i = \mu_p \tag{7}$$

$$y_t \geq \mu_p - \sum_{i=1}^{n}r_{it}x_i \qquad t = 1, 2\ldots, T \tag{8}$$

$$y_t \geq \sum_{i=1}^{n} r_{it} x_i - \mu_p \qquad\qquad t = 1, 2 \ldots, \mathrm{T} \qquad\qquad (9)$$

### 3.3 SAD risk measure

SAD is an alternative risk metric that quantifies the downside risk of an investment return. This type of risk measurement is similar to the absolute deviation risk measure, but also incorporates a threshold or target return level, focusing on deviations below the target level. As opposed to the absolute deviation risk measure that takes into account both positive and negative deviations from the mean, the SAD risk measure focuses specifically on the downside risk or potential losses below the threshold return level [18]. Investors with a primary focus on downside protection can use it to measure risk more precisely. Based on a universe of N assets and T historical periods (past horizon time-length), the objective functions can be found in the following formulas and represented in Eqs 10 to 15. Eq 10 expresses the objective function of risk minimization and Eq 11 explains the objective function of profit maximization. Eqs 13 and 14 are used for linearization of Eq 5. To transform the risk objective function, which includes an absolute value, into a linear form, a variable $y_t$ is utilized.

$$Min\ Z_1 = \frac{1}{T} \sum_{i=1}^{T} \left| \sum_{i=1}^{N} (R_{it} - \bar{R}_I) \right| = Min\ Z_1 \frac{1}{T} \sum_{i=1}^{T} y_t \qquad\qquad (10)$$

$$Max\ Z_2 = \sum_{I=1}^{N} x_i \bar{R}_I \qquad\qquad (11)$$

$$\sum_{i=1}^{n} \mu_i x_i = \mu_p \qquad\qquad (12)$$

$$y_t \geq \mu_p - \sum_{i=1}^{n} R_{it} x_i \qquad\qquad t = 1, 2 \ldots, \mathrm{T} \qquad\qquad (13)$$

$$y_t \geq \sum_{i=1}^{n} R_{it} x_i - \mu_p \qquad\qquad t = 1, 2 \ldots, \mathrm{T} \qquad\qquad (14)$$

$$y_t \geq 0 \qquad\qquad (15)$$

### 3.4. Constraints

To improve the realistic aspect of a portfolio optimization model, additional constraints are usually incorporated. As a part of the study, the following additional practical constraints are examined to determine whether the portfolio optimization model proposed is feasible.

**3.4.1 Budget constraints.** In this constraint with the name of the budget limit, the total deduction of budgets allocated to different stocks should equal 1. These constraints are shown in Eq 16.

$$\sum_{I=1}^{N} x_i = 1 \qquad\qquad (16)$$

**3.4.2. Short-selling constraints.** According to Eq 6, the investment percentage per share can never be negative. These constraints are shown in Eq 17.

$$x_i \geq 0 \qquad\qquad i = 1, 2, \ldots, N \qquad (17)$$

**3.4.3. Cardinality constraints.** The cardinality constraint establishes both a minimum and maximum limit on the number of assets to be included in the portfolio. This constraint guarantees diversification by avoiding excessive dependence on a few assets and maintaining a balanced portfolio. In this particular article, the minimum and maximum cardinality constraints specify a range of 4 to 7 distinct stocks. These constraints are shown in Eq 18. To enhance the yield per share and minimize regret and risk, it is crucial to determine the appropriate minimum and maximum weights for investing in each share. This article specifies a range of 0.01 to 0.8 for these weights. The corresponding constraint is expressed in Eq 19.

$$m \leq \sum z_i \leq M \qquad\qquad \mathrm{m} = 4, \mathrm{M} = 7 \qquad (18)$$

$$z_i\, l_i \leq x_i \leq z_i\, u_i \qquad (19)$$

The limitations of the study are established when a stock has been out of the market for a long time or, in other words, closed, and then we want to include it in the selected stocks.

## 3.5 A proposed model incorporating the regret-based approach

It will be required to initially clarify the concept of the regret approach before explaining the mathematical concept of it. Regret is the difference between the profit or benefit obtained and the profit or advantage that could have been obtained if we knew which of those scenarios would occur. According to Stetman and Shefrin [54], regret avoidance refers to the act of avoiding regretful feelings and actions, which can lead to indecisiveness and a reluctance to make necessary changes in poor investment decisions. This behavior can cause investors to remain committed to losing positions for an extended period, rather than accepting their losses and moving on. The authors suggest that this behavior stems from a desire to avoid making mistakes and experiencing financial losses.

The theory of regret avoidance was proposed as an alternative to the expected utility theory, following a series of experimental studies. This theory suggests that when individuals make decisions and choose between two options, they not only consider the potential benefits of the chosen option but also take into account the potential benefits of the alternative option that they did not choose. This is because people are sensitive to the potential losses and costs associated with their choices.

For this study, a stochastic programming approach is employed to consider parameters with existing uncertainty. A scenario-based stochastic programming approach and a minimum-maximum relative regret approach are combined in this method. Furthermore, employing this methodology allows us to consider the accumulated regret weekly, thus enabling the determination of the likelihood of achieving an optimal stock portfolio. This, in turn, facilitates the identification of opportune weeks for investment exited to attain portfolio optimization. Consequently, this approach demonstrates the probabilistic occurrence of various scenarios each week, effectively minimizing risk measures while maximizing expected returns to minimize regrets.

These reasons justify the selection of stochastic programming as the method of choice:

i. Researchers and investors will understand the simplicity of the model's structure.

ii. Decision-makers can alter and observe the effects of maximum relative regret on their final decision.

iii. As a result of this approach, it is shown that the differences between the best value of the objective function over all scenarios and the optimal value of the same objective function over all scenarios are the greatest.

iv. By considering risk and return on investment, this method adequately addresses the uncertainty of the model based on the parameters and their value.

v. As a result of incorporating maximum relative regret, which represents the risk associated with the occurrence of each scenario, a better solution is produced for the final solution.

After applying the stochastic programming approach, the mathematical model is as follows:

$$Minimize \ \sum_{s}(\pi_s \times Z1_s) + \eta1 \times \max_{s \epsilon S}\left(\frac{Z1_s^* - Z1_s}{Z1_s^*}\right) \tag{20}$$

$$Maximize \ \sum_{s}(\pi_s \times Z2_s) - \eta2 \times \max_{s \epsilon S}\left(\frac{Z2_s^* - Z2_s}{Z2_s^*}\right) \tag{21}$$

$$Z1_s, Z1_s^*, Z2_s, Z2_s^* \geq 0 \qquad\qquad \forall S \tag{22}$$

As illustrated in Eq 18, the amount of risk is calculated based on each series of volatility risk measures, which minimizes the impact of risk as well as the maximum difference between the best value of the objective function in each case and the optimal value of the objective function in every case. As shown in Eq 19, the first part shows the total expected portfolio return under different scenarios while the second part represents the risk associated with multiple outcomes. Based on this equation, the expected total portfolio return is maximized and the maximum difference between the optimal value of the objective function and the best value of the objective function is minimized. According to Model 1, Eq 20 determines the characteristics of stochastic programming decision variables. Due to the existing max function in Eqs 18 and 19, they are nonlinear and should be linearized. Let's use "REG1" to represent the level of regret in the risk objective function, and "REG2" to represent the level of regret in the return objective function. As a result of linearizing these two equations, the mathematical model is as follows

Model 1:

$$Minimize \sum_{s}(\pi_s \times Z1_s) + \eta1 \times REG1 \tag{23}$$

$$Maximize \sum_{s}(\pi_s \times Z2_s) - \eta2 \times REG2 \tag{24}$$

S.t.

$$\frac{Z1_s^* - Z1_s}{Z1_s^*} \leq REG1 \tag{25}$$

$$\frac{Z2_s^* - Z2_s}{Z2_s^*} \leq REG2 \tag{26}$$

$$\sum_{I=1}^{N} x_i = 1 \tag{27}$$

$$x_i \geq 0 \qquad\qquad i = 1, 2, \ldots, N \tag{28}$$

$$m \leq \sum z_i \leq M \qquad\qquad \mathrm{m} = 4, \mathrm{M} = 7 \tag{29}$$

$$z_i\, l_i \leq x_i \leq z_i\, u_i \tag{30}$$

$$Z1_s, Z1_s^*, Z2_s, Z2_s^* \geq 0 \tag{31}$$

$$\text{Model 2}: \tag{32}$$

$$min\ REGT = minREG1 + maxREG2 \qquad\qquad 0 \leq REGT \leq 1 \tag{33}$$

To linearize these equations, we introduce a new variable, REGT, which shows the maximum relative regret of the objective functions 1 and 2. Using Eqs 23 and 24, we can determine the value of these variables. According to these equations, REGT should be equal to the maximum deviation from the optimal value of the objective function. Model 2 explains the objective function of REGT.

### 3.6. Augmented ε-constraint method

The goal of this section is to solve two-objective research models using the augmented epsilon-constraint method (AEC). first, explain the concept of this method and then describe the steps used to solve the model in this method.

**3.6.1 The concept of Augmented ε-constraint method.** AEC is a mathematical programming procedure that is commonly used to solve multi-objective optimization problems, such as portfolio optimization. It is especially useful for resolving multiple conflicting objectives. The objectives of portfolio optimization often include balancing risk with return. For instance, an investor may wish to maximize return while minimizing risk at the same time. Generally, higher potential returns come with greater risk associated with them-therefore, these objectives are often in conflict. The AEC method provides a method for systematically exploring these tradeoffs. It converts a multi-objective optimization problem into a series of single-objective optimization problems. Using the epsilon constraint method, one of the objectives is selected as the primary objective to optimize, while the other objectives are transformed into constraints with a specified limit (epsilon). Using this technique, we can more evenly distribute the solutions obtained along the Pareto frontier (set of optimal trade-off solutions), thereby increasing the chances of finding true Pareto-optimal solutions when constraints are not convex [55–58].

**3.6.2 The solution approach of Augmented ε-constraint metho.** Generally, multi-objective decision-making programming (MODM) is expressed as follows:

$$\begin{cases} Min\ (f_1(x), f_2(x), \ldots, f_n(x)) \\ \qquad\qquad x \in X \end{cases} \tag{34}$$

Considering the importance of risk and its application, the main goal of this research is risk,

while other goals are limited to the upper limit of epsilon and are applied within the constraints of the problem. By using the AEC method, this single-objective model is generated:

$$\begin{cases} Min\, f_1(x) \\ f_i(x) \geq e_i\, i = 2, 3, .., n \\ \quad x \in X \end{cases} \tag{35}$$

This model considers the main goal to be the first goal, and the second to nth goals to be restricted by $e_i$ maximum value. Due to the goal of efficiency for portfolio return being maximization, the clause related to this goal is defined as $f_i(x) \geq e_i$ in Eq 2.

By changing the $e_i$ values in the epsilon constraint method, different solutions are obtained, which are generally either inefficient or at least not inefficient. According to the AEC method, the range of acceptable $e_i$ changes must be determined first, and then the volatile front should be determined for different values of $e_i$. Below are explanations of these two steps:

1. the range of acceptable $e_i$ changes

To determine the interval for $e_i$ associated with the second goal, we solve the following optimization problems for every goal (j = 1,2)

$$\begin{aligned} PayOff_{jj} &= Min\, f_j(x) \\ x &\in X \end{aligned} \tag{36}$$

As a result, $x^{j,*}$ represents the optimal solution, while $PayOff_{jj} = f_j(x^{j,*})$ represents the optimal objective value. By considering each target j = 1,2 in an optimal state, we can calculate the optimal value of target i.

$$\begin{aligned} PayOff_{ij} &= Min\, f_i(x) \\ f_j(x) &= PayOff_{jj} \\ x &\in X \\ j &\neq i \end{aligned} \tag{37}$$

Now, the optimal solution $x^{i,j,*}$ has been determined based on the value of $PayOff_{jj} = f_i(x^{i,j,*})$ that is optimal for the I goal.

$$PayOff = [payOff_{ij}] \tag{38}$$

the Pareto front should be determined for different values of $e_i$

$$Min(f_i) = Min_j\{payOff_{ij}\} = payOff_{ii} \tag{39}$$

$$Max(f_i) = Max_j\{payOff_{ij}\} \tag{40}$$

$$R(f_i) = Max(f_i) - Min(f_i) \tag{41}$$

According to the above definition, $e_i$ should fall between $Max(f_i)$ and $Min(f_i)$ and $R(f_i)$ should be used to normalize the objectives in the AEC objective function.

2. developing with the AEC method

An AEC planning model has been developed to resolve a problem in the method:

$$\begin{cases} Min\, f_1(x) - \sum_{i=2}^{n} \phi_i s_i \\ f_i(x) + s_i = e_i \; i = 2, 3, .., n \\ \qquad x \in X \\ \qquad s_i \geq 0 \end{cases} \tag{42}$$

### 3.7. fundamental analysis

The fundamental analysis is a process used to evaluate a security or investment's intrinsic value by analyzing various factors associated with the underlying asset, including financial statements, industry trends, management quality, competitive advantages, and macroeconomic factors, among others. A fundamental strength analysis is an evaluation of an investment's future performance based on the fundamental strength of an investment[59].

By considering economic and financial indicators, industry conditions, and specific company characteristics, fundamental analysis assists analysts in assessing the value of a security. After carefully examining the stock trends in the market and also examining all the fundamental criteria, considering that these criteria were effective in the stock trends, we selected them in consultation with the experts in this field. In this article, we will present a list of the most important criteria that are essential for a fundamental analysis to be effective[60–65]:

❖ Amount of earnings; An investor may view a company with a history of increasing earnings as a potential investment opportunity when they view earnings per share (EPS) revenue and net income measures.

❖ Earnings; This includes measures, like Earnings Per Share (EPS) revenue and net income. Investors often view companies with a track record of increasing earnings as investment opportunities.

❖ Valuation Ratios; These ratios consist of Price/Earnings (P/E) Price/Book (P/B) Price/Sales (P/S) and Price/Cash Flow (P/CF). They assist investors in determining whether a stock is overvalued or undervalued.

❖ Dividends; For investors seeking income the dividend yield and history of dividend payments play a role in decision-making.

❖ Cash Flow; By analyzing cash flow operating cash flow well as cash flow from investing and financing activities one can gain insights into the financial well-being of the company and its ability to generate cash.

❖ Management Quality; The expertise, competence, and integrity demonstrated by a company's management team can have an impact, on its performance.

❖ Competitive Advantage; Companies that possess a position or "moat" are generally considered promising investments.

❖ Current Trends, in the Industry; The company's position within its industry the overall well-being of the industry, and emerging patterns that affect the industry have the potential to influence the returns on a stock.

❖ Influence of Macroeconomic Factors; Interest rates, inflation, GDP growth, unemployment rates, and other macroeconomic factors can have an impact on both a company's performance and the broader stock market.

❖ Assessing Growth Opportunities; Evaluating a company's potential for growth involves considering factors such as new product developments, expansion into new markets, and overall trends, within the industry.

After careful evaluation of the aforementioned criteria, appropriate stocks from diverse industries have been chosen based on fundamental analysis metrics.

## 4. Case study

For data analysis, we used the 20 stocks of New York that were selected by fundamental analysis factors extracted from Yahoo Finance's historical data, which represents significant leaders in the world's industries based on their capitalization. In this portfolio, 20 stocks are represented by eight notable industries. The mid-return of each selected stock in 3 scenarios is shown in Fig 1. In this paper, we demonstrate the evolution of return and risk using three scenarios of short-term, mid-term, and long-term investment scenarios. In the absence of actual decision-makers, we create 3 scenarios for the return and the risk as follows: We used historical data from 20, 50, and 100 weeks that included the different horizons of investment, and extracted the average return and risk measures including the SV, MAD, and SAD. As a result, Scenario 1, which corresponds to 20 weeks past the horizon, represents short-term behavior, Scenario 2, represents mid-term behavior, and Scenario 3, represents long-term behavior. Fig 2 shows the average return of each selected stock in every scenario.

## 5. Computational results

In this section, we use historical data from the New York Stock Exchange (NY). A total of 100 weeks and 150 chosen stocks historical data from 13 September 2021 to 7 August 2023 are covered by the data weekly. the proposed model is run by GAMS 24.1.2 and solved with Cplex and Baron solver. The results show the selected stocks. The descriptive statistics of the selected assets are presented in Table 3. The solution method of research models is exact. It is worth mentioning that the model type in the MAD and SAD risk measures is MIP type and the mode type in the SV measure is MINLP type.

the solution time of the proposed models (SV, MAD, and SAD) in larger dimensions is explained in Table 4.

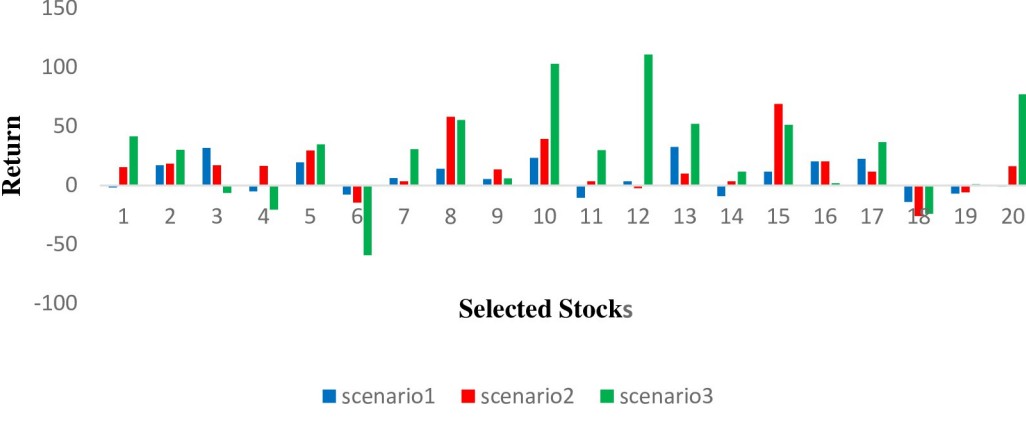

**Fig 2. Return of Each selected stock by fundamental analysis.**

**Table 3. Descriptive statistics of the selected assets.**

| stocks | name | mean | variance | sd | max | min | stocks | name | mean | variance | sd | max | min |
|---|---|---|---|---|---|---|---|---|---|---|---|---|---|
| 1 | ABBV | 0.004 | 0.001 | 0.031 | 0.102 | -0.078 | 76 | FYBR | -0.001 | 0.004 | 0.066 | 0.202 | -0.213 |
| 2 | AMGN | 0.003 | 0.001 | 0.031 | 0.097 | -0.069 | 77 | GE | 0.005 | 0.002 | 0.046 | 0.115 | -0.163 |
| 3 | AMZN | -0.001 | 0.003 | 0.054 | 0.140 | -0.139 | 78 | GILD | 0.003 | 0.001 | 0.033 | 0.169 | -0.072 |
| 4 | BABA | -0.002 | 0.007 | 0.081 | 0.249 | -0.160 | 79 | GL | 0.004 | 0.001 | 0.032 | 0.077 | -0.078 |
| 5 | BRK-B | 0.004 | 0.001 | 0.028 | 0.077 | -0.081 | 80 | GM | -0.001 | 0.003 | 0.056 | 0.110 | -0.128 |
| 6 | BUD | 0.001 | 0.001 | 0.036 | 0.089 | -0.154 | 81 | GOOG | 0.001 | 0.002 | 0.048 | 0.126 | -0.102 |
| 7 | CMCSA | -0.001 | 0.001 | 0.038 | 0.097 | -0.124 | 82 | HD | 0.001 | 0.001 | 0.038 | 0.109 | -0.088 |
| 8 | DASH | -0.006 | 0.007 | 0.086 | 0.238 | -0.186 | 83 | IAK | -0.004 | 0.004 | 0.060 | 0.129 | -0.179 |
| 9 | DIS | -0.006 | 0.002 | 0.044 | 0.140 | -0.096 | 84 | IBKR | 0.005 | 0.002 | 0.047 | 0.121 | -0.144 |
| 10 | ELV | 0.003 | 0.001 | 0.037 | 0.103 | -0.080 | 85 | IFF | -0.005 | 0.003 | 0.054 | 0.121 | -0.195 |
| 11 | FCX | 0.006 | 0.004 | 0.067 | 0.159 | -0.159 | 86 | IPG | 0.000 | 0.002 | 0.042 | 0.101 | -0.164 |
| 12 | HDB | 0.001 | 0.002 | 0.041 | 0.106 | -0.140 | 87 | ISRG | 0.001 | 0.003 | 0.053 | 0.194 | -0.124 |
| 13 | HES | 0.010 | 0.004 | 0.061 | 0.176 | -0.195 | 88 | ITW | 0.003 | 0.001 | 0.035 | 0.094 | -0.094 |
| 14 | HSY | 0.003 | 0.001 | 0.027 | 0.080 | -0.084 | 89 | JEF | 0.002 | 0.002 | 0.047 | 0.102 | -0.116 |
| 15 | JD | -0.003 | 0.007 | 0.086 | 0.357 | -0.245 | 90 | JNJ | 0.002 | 0.001 | 0.023 | 0.076 | -0.038 |
| 16 | KMB | 0.000 | 0.001 | 0.030 | 0.104 | -0.090 | 91 | JPM | 0.001 | 0.001 | 0.038 | 0.119 | -0.092 |
| 17 | LIN | 0.003 | 0.001 | 0.033 | 0.110 | -0.065 | 92 | KNSL | 0.010 | 0.003 | 0.050 | 0.152 | -0.127 |
| 18 | NFLX | 0.000 | 0.005 | 0.074 | 0.259 | -0.368 | 93 | KO | 0.002 | 0.001 | 0.025 | 0.086 | -0.074 |
| 19 | NKE | -0.002 | 0.002 | 0.044 | 0.108 | -0.143 | 94 | LAD | 0.001 | 0.003 | 0.055 | 0.206 | -0.148 |
| 20 | NVO | 0.010 | 0.003 | 0.053 | 0.327 | -0.111 | 95 | LNG | 0.009 | 0.003 | 0.052 | 0.216 | -0.093 |
| 21 | OXY | 0.011 | 0.005 | 0.072 | 0.449 | -0.128 | 96 | LPLA | 0.006 | 0.003 | 0.054 | 0.154 | -0.156 |
| 22 | PBR-A | 0.005 | 0.005 | 0.069 | 0.231 | -0.166 | 97 | LYB | 0.002 | 0.002 | 0.043 | 0.085 | -0.125 |
| 23 | RTX | 0.002 | 0.001 | 0.035 | 0.118 | -0.094 | 98 | MA | 0.002 | 0.001 | 0.036 | 0.090 | -0.104 |
| 24 | SCCO | 0.005 | 0.003 | 0.056 | 0.143 | -0.123 | 99 | MAR | 0.005 | 0.002 | 0.044 | 0.112 | -0.113 |
| 25 | SHW | 0.000 | 0.002 | 0.043 | 0.123 | -0.100 | 100 | MCD | 0.003 | 0.001 | 0.025 | 0.078 | -0.062 |
| 26 | TD | 0.000 | 0.001 | 0.030 | 0.059 | -0.075 | 101 | MCK | 0.008 | 0.001 | 0.031 | 0.079 | -0.102 |
| 27 | TMUS | 0.001 | 0.001 | 0.034 | 0.113 | -0.071 | 102 | MDT | -0.003 | 0.001 | 0.033 | 0.065 | -0.105 |
| 28 | TSLA | 0.004 | 0.008 | 0.088 | 0.333 | -0.180 | 103 | META | 0.001 | 0.005 | 0.071 | 0.245 | -0.237 |
| 29 | TU | -0.002 | 0.001 | 0.026 | 0.055 | -0.073 | 104 | MHK | -0.004 | 0.003 | 0.056 | 0.220 | -0.122 |
| 30 | UPS | 0.000 | 0.001 | 0.038 | 0.134 | -0.111 | 105 | MMC | 0.004 | 0.001 | 0.032 | 0.100 | -0.063 |
| 31 | WFC | -0.001 | 0.000 | 0.015 | 0.057 | -0.037 | 106 | MORN | 0.000 | 0.002 | 0.049 | 0.102 | -0.141 |
| 32 | WMT | 0.002 | 0.001 | 0.033 | 0.078 | -0.195 | 107 | MOS | 0.004 | 0.004 | 0.067 | 0.208 | -0.141 |
| 33 | XOM | 0.008 | 0.002 | 0.045 | 0.157 | -0.143 | 108 | MRK | 0.005 | 0.001 | 0.033 | 0.106 | -0.074 |
| 34 | ABNB | 0.001 | 0.006 | 0.074 | 0.209 | -0.166 | 109 | MRO | 0.010 | 0.005 | 0.067 | 0.237 | -0.203 |
| 35 | ACGL | 0.007 | 0.001 | 0.035 | 0.175 | -0.074 | 110 | MTB | 0.001 | 0.002 | 0.049 | 0.152 | -0.136 |
| 36 | AIZ | 0.000 | 0.001 | 0.036 | 0.075 | -0.117 | 111 | MTCH | -0.009 | 0.006 | 0.074 | 0.196 | -0.161 |
| 37 | ALB | 0.002 | 0.006 | 0.077 | 0.257 | -0.174 | 112 | MUR | 0.010 | 0.005 | 0.073 | 0.239 | -0.238 |
| 38 | ALLY | -0.003 | 0.004 | 0.059 | 0.161 | -0.154 | 113 | NEM | -0.001 | 0.002 | 0.049 | 0.143 | -0.121 |
| 39 | AON | 0.001 | 0.001 | 0.034 | 0.080 | -0.105 | 114 | NFG | 0.001 | 0.001 | 0.032 | 0.098 | -0.099 |
| 40 | APD | 0.002 | 0.001 | 0.036 | 0.103 | -0.093 | 115 | NXST | 0.003 | 0.002 | 0.044 | 0.118 | -0.141 |
| 41 | APTV | -0.002 | 0.004 | 0.060 | 0.144 | -0.222 | 116 | NYT | 0.000 | 0.002 | 0.045 | 0.139 | -0.116 |
| 42 | ASH | 0.001 | 0.001 | 0.037 | 0.089 | -0.104 | 117 | OLN | 0.004 | 0.003 | 0.058 | 0.146 | -0.211 |
| 43 | AU | 0.005 | 0.005 | 0.068 | 0.230 | -0.143 | 118 | OMC | 0.002 | 0.001 | 0.038 | 0.105 | -0.130 |
| 44 | AVTR | -0.002 | 0.004 | 0.062 | 0.401 | -0.153 | 119 | ORI | 0.003 | 0.001 | 0.032 | 0.085 | -0.085 |
| 45 | BAC | -0.001 | 0.002 | 0.043 | 0.105 | -0.114 | 120 | PBF | 0.019 | 0.008 | 0.092 | 0.263 | -0.197 |
| 46 | BK | 0.003 | 0.002 | 0.049 | 0.307 | -0.101 | 121 | PFE | -0.001 | 0.001 | 0.038 | 0.127 | -0.094 |
| 47 | BSX | 0.002 | 0.001 | 0.031 | 0.077 | -0.067 | 122 | PG | 0.001 | 0.001 | 0.028 | 0.091 | -0.077 |

*(Continued)*

**Table 3.** (Continued)

| stocks | name | mean | variance | sd | max | min | stocks | name | mean | variance | sd | max | min |
|---|---|---|---|---|---|---|---|---|---|---|---|---|---|
| 48 | BX | 0.001 | 0.004 | 0.064 | 0.208 | -0.161 | 123 | PKG | 0.002 | 0.001 | 0.035 | 0.113 | -0.154 |
| 49 | CBOE | 0.002 | 0.001 | 0.027 | 0.058 | -0.103 | 124 | PYPL | -0.013 | 0.005 | 0.068 | 0.230 | -0.229 |
| 50 | CBT | 0.005 | 0.002 | 0.046 | 0.123 | -0.155 | 125 | RGLD | 0.002 | 0.002 | 0.041 | 0.109 | -0.093 |
| 51 | CHRD | 0.008 | 0.004 | 0.060 | 0.139 | -0.230 | 126 | RJF | 0.003 | 0.002 | 0.046 | 0.181 | -0.121 |
| 52 | CHTR | -0.003 | 0.003 | 0.058 | 0.131 | -0.199 | 127 | RKT | -0.001 | 0.006 | 0.079 | 0.288 | -0.224 |
| 53 | CINF | 0.001 | 0.002 | 0.040 | 0.103 | -0.122 | 128 | ROKU | -0.009 | 0.012 | 0.108 | 0.303 | -0.314 |
| 54 | CLF | 0.002 | 0.007 | 0.084 | 0.206 | -0.230 | 129 | ROL | 0.001 | 0.002 | 0.039 | 0.149 | -0.104 |
| 55 | CMC | 0.008 | 0.002 | 0.048 | 0.129 | -0.118 | 130 | RY | 0.000 | 0.001 | 0.026 | 0.056 | -0.074 |
| 56 | COF | -0.002 | 0.003 | 0.053 | 0.134 | -0.149 | 131 | SBUX | -0.001 | 0.002 | 0.041 | 0.102 | -0.109 |
| 57 | CPNG | -0.001 | 0.006 | 0.075 | 0.173 | -0.183 | 132 | SIRI | 0.000 | 0.005 | 0.069 | 0.491 | -0.278 |
| 58 | CRC | 0.005 | 0.003 | 0.058 | 0.138 | -0.174 | 133 | SPOT | -0.003 | 0.004 | 0.067 | 0.185 | -0.191 |
| 59 | CSX | 0.002 | 0.001 | 0.038 | 0.093 | -0.096 | 134 | STE | 0.002 | 0.002 | 0.045 | 0.115 | -0.136 |
| 60 | CTVA | 0.004 | 0.001 | 0.036 | 0.103 | -0.082 | 135 | STLD | 0.008 | 0.004 | 0.065 | 0.190 | -0.157 |
| 61 | CVX | 0.007 | 0.002 | 0.043 | 0.136 | -0.154 | 136 | SWN | 0.008 | 0.007 | 0.084 | 0.315 | -0.262 |
| 62 | DFS | -0.001 | 0.002 | 0.048 | 0.116 | -0.113 | 137 | SYF | -0.002 | 0.003 | 0.052 | 0.149 | -0.137 |
| 63 | DG | -0.001 | 0.002 | 0.048 | 0.217 | -0.193 | 138 | TFC | -0.003 | 0.002 | 0.049 | 0.110 | -0.213 |
| 64 | DKNG | 0.003 | 0.013 | 0.116 | 0.316 | -0.259 | 139 | TKO | 0.008 | 0.002 | 0.043 | 0.230 | -0.066 |
| 65 | DLTR | 0.008 | 0.003 | 0.057 | 0.290 | -0.198 | 140 | TROW | -0.005 | 0.003 | 0.055 | 0.297 | -0.115 |
| 66 | EA | 0.001 | 0.001 | 0.038 | 0.105 | -0.116 | 141 | TRV | 0.001 | 0.001 | 0.029 | 0.079 | -0.076 |
| 67 | EBAY | -0.003 | 0.002 | 0.048 | 0.161 | -0.101 | 142 | TSN | -0.002 | 0.001 | 0.038 | 0.110 | -0.195 |
| 68 | ECL | 0.001 | 0.002 | 0.044 | 0.155 | -0.146 | 143 | UNP | 0.001 | 0.001 | 0.035 | 0.106 | -0.086 |
| 69 | EMN | -0.001 | 0.002 | 0.044 | 0.115 | -0.155 | 144 | V | 0.001 | 0.001 | 0.036 | 0.114 | -0.087 |
| 70 | EPD | 0.004 | 0.001 | 0.033 | 0.083 | -0.151 | 145 | WBS | 0.000 | 0.002 | 0.050 | 0.131 | -0.173 |
| 71 | EQT | 0.011 | 0.005 | 0.068 | 0.266 | -0.251 | 146 | WFRD | 0.022 | 0.008 | 0.092 | 0.327 | -0.200 |
| 72 | ET | 0.005 | 0.002 | 0.041 | 0.123 | -0.155 | 147 | WRB | 0.003 | 0.001 | 0.029 | 0.066 | -0.072 |
| 73 | EXPE | -0.001 | 0.004 | 0.064 | 0.154 | -0.243 | 148 | WTW | -0.001 | 0.001 | 0.031 | 0.082 | -0.105 |
| 74 | FOXA | 0.000 | 0.001 | 0.036 | 0.095 | -0.081 | 149 | XP | -0.003 | 0.006 | 0.080 | 0.281 | -0.182 |
| 75 | FWONA | 0.005 | 0.002 | 0.041 | 0.123 | -0.092 | 150 | YUM | 0.000 | 0.001 | 0.027 | 0.072 | -0.061 |

The results' payoff tables present the returns and associated risks of the stock portfolio across different investment scenarios, namely, 1, 2, and 3. Tables 5–7 illustrate the three-payoff table for SV, MAD, and SAD risk measures respectively.

**Table 4. The solution time of the proposed models.**

| Volatility risk measures | scenarios | Solution time (min: seconds) |
|---|---|---|
| SV | 1 | 07:13 |
| | 2 | 06:24 |
| | 3 | 17:03 |
| MAD | 1 | 09:14 |
| | 2 | 10:33 |
| | 3 | 13:21 |
| SAD | 1 | 03:52 |
| | 2 | 10:55 |
| | 3 | 12:48 |

**Table 5. The payoff table of SV risk measure in the 3 scenarios.**

| objectives | Scenario 1 | | Scenario 2 | | Scenario 3 | |
|---|---|---|---|---|---|---|
| | SV | Return | SV | Return | SV | Return |
| Min SV | 0.002 | 0.253 | 0.001 | 0.451 | 0.0002 | 0.712 |
| Max Return | 0.018 | 0.07 | 0.023 | 0.025 | 0.031 | 0.019 |

**Table 6. The payoff table of MAD risk measure in the 3 scenarios.**

| objectives | Scenario 1 | | Scenario 2 | | Scenario 3 | |
|---|---|---|---|---|---|---|
| | MAD | Return | MAD | Return | MAD | Return |
| Min MAD | 0.005 | 0.412 | 0.004 | 0.673 | 0.003 | 0.844 |
| Max return | 0.004 | 0.027 | 0.001 | 0.015 | 0.002 | 0.012 |

**Table 7. The payoff table of SAD risk measure in the 3 scenarios.**

| objectives | Scenario 1 | | Scenario 2 | | Scenario 3 | |
|---|---|---|---|---|---|---|
| | SAD | Return | SAD | Return | SAD | Return |
| Min SAD | 0.001 | 0.382 | 0.002 | 0.516 | 0.0015 | 0.733 |
| Max Return | 0.077 | 0.018 | 0.123 | 0.12 | 0.199 | 0.012 |

After completing the final model using regret-based analysis, we have determined the probabilities of the optimal stock portfolio being realized in different investment scenarios, specifically for the 20, 50, and 100-week periods. These probabilities are calculated for each investment week and can be found in Tables 8–10. These values represent the influence of the probability of each investment week on the composition of the optimal portfolio in each investment scenario.

In Fig 3, the three Pareto fronts represent distinct scenarios, each associated with different returns. These fronts display dissimilarity due to their corresponding return scenarios. Notably, scenario 3 yields higher returns compared to scenarios 1 and 2, particularly evident in the upper right region of the chart where maximum returns are achieved. The reviewer's insight suggests that in this upper right region, focused on maximizing returns, there is a greater level of dispersion compared to the lower left region where risk minimization is prioritized. This finding may indicate why the more resilient areas are concentrated in the latter region, suggesting a relationship between risk management and robust performance.

**Table 8. Probability of 20-week scenario ($\pi_s$) for each week.**

| Weeks | $\pi_s$ | weeks | $\pi_s$ |
|---|---|---|---|
| 1 | 0.015 | 11 | 0.06 |
| 2 | 0.046 | 12 | 0.031 |
| 3 | 0.03 | 13 | 0.07 |
| 4 | 0.024 | 14 | 0.02 |
| 5 | 0.023 | 15 | 0.015 |
| 6 | 0.018 | 16 | 0.034 |
| 7 | 0.021 | 17 | 0.018 |
| 8 | 0.048 | 18 | 0.021 |
| 9 | 0.012 | 19 | 0.035 |
| 10 | 0.05 | 20 | 0.028 |

**Table 9. The probability of a 50-week scenario ($\pi_s$) for each week.**

| weeks | $\pi_s$ | weeks | $\pi_s$ | weeks | $\pi_s$ | weeks | $\pi_s$ | weeks | $\pi_s$ |
|---|---|---|---|---|---|---|---|---|---|
| 1 | 0.004 | 11 | 0.004 | 21 | 0.009 | 31 | 0.005 | 41 | 0.014 |
| 2 | 0.02 | 12 | 0.012 | 22 | 0.011 | 32 | 0.012 | 42 | 0.003 |
| 3 | 0.011 | 13 | 0.05 | 23 | 0.007 | 33 | 0.007 | 43 | 0.007 |
| 4 | 0.012 | 14 | 0.013 | 24 | 0.008 | 34 | 0.018 | 44 | 0.008 |
| 5 | 0.008 | 15 | 0.004 | 25 | 0.015 | 35 | 0.008 | 45 | 0.008 |
| 6 | 0.006 | 16 | 0.016 | 26 | 0.014 | 36 | 0.009 | 46 | 0.009 |
| 7 | 0.012 | 17 | 0.005 | 27 | 0.011 | 37 | 0.014 | 47 | 0.012 |
| 8 | 0.017 | 18 | 0.01 | 28 | 0.013 | 38 | 0.016 | 48 | 0.011 |
| 9 | 0.006 | 19 | 0.013 | 29 | 0.018 | 39 | 0.014 | 49 | 0.014 |
| 10 | 0.011 | 20 | 0.014 | 30 | 0.012 | 40 | 0.011 | 50 | 0.013 |

In Table 11, we present the optimal weights assigned to the selected stocks for each risk indicator across three different scenarios. Additionally, the table provides information on the lowest levels of total regret (REGT) and risk incurred for each scenario, as well as the number of optimal effective stocks out of the 20 selected stocks based on fundamental analysis. The findings demonstrate that as the investment period increases and scenario 3 (long-term) is chosen concerning each risk measure, the amount of REGT decreases while the total return increases. It is worth noting that the total regret value falls within the range of 0 to 1. The closer the value is to zero, the lower the level of regret and the greater the robustness of the outcomes. The bold results in the table highlight the corresponding explanations provided.

Once the model is solved, the allocation of weights to individual stocks in each scenario is quantitatively expressed on a scale from 0 to 1. It is important to note that the weighting of each stock in a scenario may vary depending on the investment horizon and risk parameters.

**Table 10. Probability of 100-week scenario ($\pi_s$) for each week.**

| weeks | $\pi_s$ | Weeks | $\pi_s$ | weeks | $\pi_s$ | weeks | $\pi_s$ | weeks | $\pi_s$ |
|---|---|---|---|---|---|---|---|---|---|
| 1 | 0.004 | 21 | 0.003 | 41 | 0.002 | 61 | 0.002 | 81 | 0.005 |
| 2 | 0.008 | 22 | 0.007 | 42 | 0.006 | 62 | 0.004 | 82 | 0.003 |
| 3 | 0.009 | 23 | 0.003 | 43 | 0.005 | 63 | 0.007 | 83 | 0.002 |
| 4 | 0.007 | 24 | 0.003 | 44 | 0.002 | 64 | 0.007 | 84 | 0.006 |
| 5 | 0.004 | 25 | 0.007 | 45 | 0.004 | 65 | 0.005 | 85 | 0.004 |
| 6 | 0.002 | 26 | 0.009 | 46 | 0.004 | 66 | 0.002 | 86 | 0.002 |
| 7 | 0.005 | 27 | 0.004 | 47 | 0.009 | 67 | 0.005 | 87 | 0.005 |
| 8 | 0.011 | 28 | 0.007 | 48 | 0.008 | 68 | 0.007 | 88 | 0.005 |
| 9 | 0.002 | 29 | 0.008 | 49 | 0.008 | 69 | 0.006 | 89 | 0.005 |
| 10 | 0.007 | 30 | 0.004 | 50 | 0.008 | 70 | 0.004 | 90 | 0.008 |
| 11 | 0.01 | 31 | 0.003 | 51 | 0.007 | 71 | 0.002 | 91 | 0.008 |
| 12 | 0.004 | 32 | 0.006 | 52 | 0.008 | 72 | 0.006 | 92 | 0.006 |
| 13 | 0.01 | 33 | 0.003 | 53 | 0.007 | 73 | 0.005 | 93 | 0.006 |
| 14 | 0.005 | 34 | 0.009 | 54 | 0.002 | 74 | 0.007 | 94 | 0.008 |
| 15 | 0.009 | 35 | 0.004 | 55 | 0.005 | 75 | 0.005 | 95 | 0.005 |
| 16 | 0.007 | 36 | 0.004 | 56 | 0.006 | 76 | 0.008 | 96 | 0.008 |
| 17 | 0.003 | 37 | 0.007 | 57 | 0.01 | 77 | 0.004 | 97 | 0.005 |
| 18 | 0.002 | 38 | 0.008 | 58 | 0.006 | 78 | 0.003 | 98 | 0.007 |
| 19 | 0.009 | 39 | 0.008 | 59 | 0.003 | 79 | 0.008 | 99 | 0.003 |
| 20 | 0.005 | 40 | 0.005 | 60 | 0.007 | 80 | 0.003 | 100 | 0.006 |

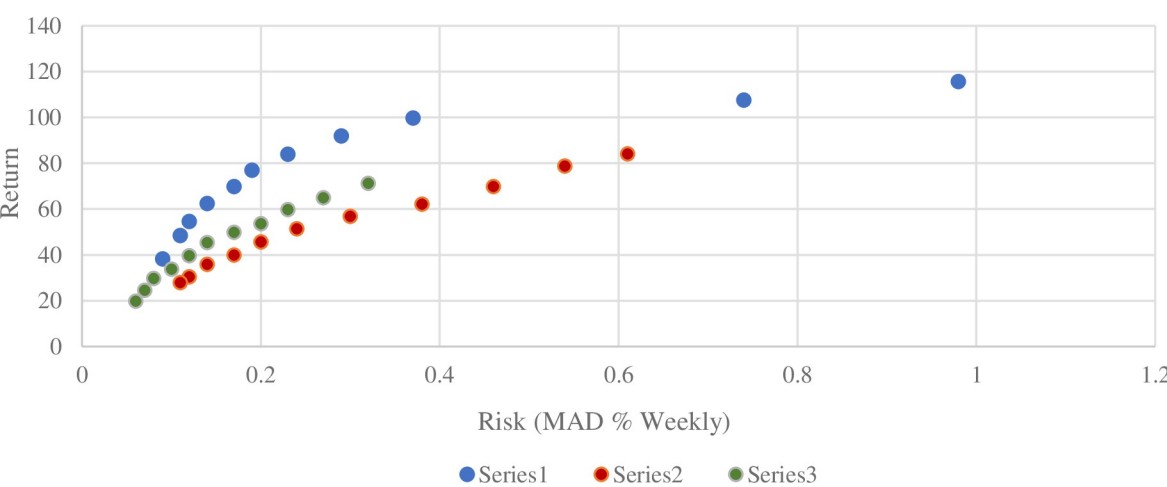

**Fig 3. Three Efficient Frontiers for Scenario in MAD.**

The novelty lies in examining the connection between risk measures and the level of regret. The findings suggest that as the investment horizon increases, indicating a higher risk appetite among investors, the level of regret decreases alongside the increase in profits. The results obtained demonstrate a decrease in regret as the investment period increases for risk measures considered. Among them, the MAD risk measure exhibits the lowest value of 0.104, indicating the least amount of regret. Therefore, for investing in this portfolio, the mad risk measure with an investment period of 100 weeks is the most suitable option.

## 6. Discussion

The empirical analysis conducted in this study offers compelling insights into the application of regret-based optimization for managing stock portfolios. Our analysis involved assigning different weights to mean absolute deviation risk measures, enabling us to select the most appropriate risk scenarios and measures. By minimizing regret and missed opportunities, our goal was to enhance portfolio performance and improve risk management capabilities.

**Table 11. Details of the obtained solutions for 3 risk measures in 3 scenarios.**

| Volatility risk measures | S | REGT | Return | S.P | Optimal selection |
|---|---|---|---|---|---|
| SV | 1 | 0.343 | 58.33 | 4 | $(x_{85} = 0.042, x_{113} = 0.355, x_{121} = 0434, x_{149} = 0.169)$ |
| | 2 | 0.297 | 64.92 | 4 | $(x_{15} = 0.061, x_{63} = 0.236, x_{121} = 0.53, x_{146} = 0.173)$ |
| | **3** | **0.145** | **71.18** | **5** | $(\mathbf{x_{85} = 0.011, x_{113} = 0.047, x_{124} = 0.002, x_{125} = 0.771, x_{149} = 0.169})$ |
| MAD | 1 | 0.299 | 42.12 | 6 | $(x_{3} = 0.192, x_{7} = 0.115, x_{31} = 0.33, x_{33} = 0.086, x_{98} = 0.141, x_{143} = 0.136)$ |
| | 2 | 0.166 | 99.87 | 6 | $(x_{14} = 0.121, x_{18} = 0.086, x_{31} = 0.433, x_{100} = 0.089, x_{122} = 0.161, x_{142} = 0.11)$ |
| | **3** | **0.104** | **115.65** | **6** | $(\mathbf{x_{7} = 0.081, x_{14} = 0.186, x_{31} = 0.502, x_{100} = 0.116, x_{110} = 0.037, x_{121} = 0.078})$ |
| SAD | 1 | 0.325 | 26.54 | 4 | $(x_{64} = 0.408, x_{100} = 0.098, x_{101} = 0.165, x_{149} = 0.329)$ |
| | 2 | 0.311 | 40.09 | 6 | $(x_{17} = 0.213, x_{18} = 0.108, x_{20} = 0.12, x_{43} = 0.23, x_{108} = 0.123, x_{103} = 0.206)$ |
| | **3** | **0.137** | **84.15** | **6** | $(\mathbf{x_{14} = 0.18, x_{31} = 0.296, x_{33} = 0.131, x_{78} = 0.101}, x_{95} = 0.11, x_{100} = 0.182)$ |

**S** = Scenario, **REGT** = Regret total, **S.P** = Selected Portfolio in Optimize mode.

The results obtained from our study demonstrate the effectiveness of the regret-based approach in portfolio optimization. It was evident that incorporating regret as a measure of performance empowered investors to make more informed investment decisions. This regret-based optimization framework emerged as a valuable tool for investors to strike a balance between risk and return, particularly concerning volatility risk measures. Furthermore, the empirical evidence derived from the New York Stock Market substantiated the practical applicability of our approach in real-world scenarios.

However, it is vital to critically analyze our study's findings concerning the existing body of research. While no specific studies are cited in this discussion, comparing our methodology, results, and conclusions with similar studies in the literature is key. This critical analysis would provide a comprehensive understanding of the advancements made in this field and help identify potential gaps or contradictions between our findings and those of prior research.

Considering the current status quo of the research, our study contributes to the literature by showcasing the effectiveness of regret-based optimization in managing stock portfolios. However, it is important to recognize certain limitations in our study. Firstly, our analysis focused solely on the New York Stock Market, which raises concerns about the generalizability of our findings to other stock markets or regions. Future research should aim to replicate these analyses in diverse markets to determine the robustness and broader applicability of the regret-based approach.

In addition, our study primarily concentrated on volatility risk measures, warranting further investigation into the applicability of regret-based optimization with other risk measures. Different risk factors, such as liquidity risk, credit risk, or geopolitical risk, may have a significant impact on portfolio outcomes. Exploring the effectiveness of the regret-based framework in incorporating these risk measures would provide additional insights and guidance to investors.

To build upon the current status quo of research, it is crucial to compare our findings with those of other studies. Analyzing and contrasting our methodology, results, and interpretations with prior research would deepen our understanding of the field and identify any inconsistencies or gaps that need to be addressed. This critical analysis would contribute to the advancement of knowledge in regret-based portfolio optimization.

## 7. Conclusion

The increasing popularity of scenario-based portfolio optimization is evident in recent advancements in portfolio optimization research. To better understand portfolio stability and development, researchers are utilizing innovative tools and techniques. Stochastic optimization, which allows for input portfolio parameters to be considered, is not only valuable for theoretical research but also for practical investors.

In policy Recommendation, Regulatory authorities should encourage the adoption of scenario-based portfolio optimization techniques as part of risk management practices in investment institutions. Governments should collaborate with financial industry stakeholders to develop standardized guidelines and best practices for incorporating scenario-based approaches into investment decision-making processes. Education and training programs should be established to improve financial professionals' understanding and proficiency in utilizing scenario-based optimization methods.

Investors are advised to consider incorporating scenario-based optimization techniques into their portfolio management strategies. These techniques can help identify robust investment opportunities that adequately balance risk and return in the presence of uncertainties. Diversification remains a key strategy, and investors should use scenario-based approaches to

understand the potential impact of different market conditions and adjust their portfolios accordingly. Regular monitoring and periodic reassessment of investment portfolios using scenario-based models can provide valuable insights and improve decision-making.

Future research might be driven towards examining the effectiveness of the method in portfolio optimization for more objective functions and also in other multi-objective problems. In addition, other robustness models and even fuzzy methods in parameters in the same context of the regret criterion may be developed in combination with other multi-objective techniques appropriate for generating representations of the Pareto front. To further enhance the body of knowledge on stochastic portfolio optimization, future research could consider additional factors such as liquidity risk, transaction costs, and investor preferences. Moreover, incorporating machine learning techniques or exploring alternative regret-based frameworks could provide further avenues for study and also explore the effectiveness of scenario-based portfolio optimization methods in different asset classes and market environments to provide broader applicability insights for investors. Researchers should investigate the potential integration of artificial intelligence and machine learning techniques into scenario-based optimization frameworks to enhance decision-making capabilities. Furthermore, beyond addressing this particular concern, robust optimization of the data-driven stock portfolio was employed for its application in behavioral finance matters.

## Author Contributions

**Conceptualization:** Seyed Jafar Sadjadi, Emran Mohammadi.

**Data curation:** AmirMohammad Larni-Fooeik.

**Investigation:** AmirMohammad Larni-Fooeik, Seyed Jafar Sadjadi.

**Methodology:** Emran Mohammadi.

**Resources:** AmirMohammad Larni-Fooeik.

**Software:** AmirMohammad Larni-Fooeik.

**Validation:** Seyed Jafar Sadjadi.

**Visualization:** Seyed Jafar Sadjadi.

**Writing – original draft:** AmirMohammad Larni-Fooeik.

**Writing – review & editing:** AmirMohammad Larni-Fooeik, Seyed Jafar Sadjadi.

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
