## [Decision Letter · Decision Letter 0]

20 Dec 2023

PONE-D-23-39475Stochastic Portfolio Optimization: A Regret-Based Approach On Volatility Risk Measures (An Empirical Evidence From The New York Stock Market)PLOS ONE

Dear Dr. Mohammadi,

Thank you for submitting your manuscript to PLOS ONE. After careful consideration, we feel that it has merit but does not fully meet PLOS ONE’s publication criteria as it currently stands. Therefore, we invite you to submit a revised version of the manuscript that addresses the points raised during the review process.

We look forward to receiving your revised manuscript.

Kind regards,

Shazia Rehman, Ph.D.

Academic Editor

PLOS ONE

Journal Requirements:

Reviewers' comments:

Reviewer's Responses to Questions

**Comments to the Author**

1. Is the manuscript technically sound, and do the data support the conclusions?

Reviewer #1: Partly

Reviewer #2: Partly

Reviewer #3: Partly

2. Has the statistical analysis been performed appropriately and rigorously? 

Reviewer #1: N/A

Reviewer #2: No

Reviewer #3: Yes

3. Have the authors made all data underlying the findings in their manuscript fully available?

Reviewer #1: Yes

Reviewer #2: No

Reviewer #3: Yes

4. Is the manuscript presented in an intelligible fashion and written in standard English?

Reviewer #1: No

Reviewer #2: No

Reviewer #3: Yes

5. Review Comments to the Author

Reviewer #1: - The paper's contribution is quite minimal. It is recommended to enhance the quality of the paper by implementing novel methodologies in the realm of uncertain programming such as data-driven robust optimization.

- It should be explained about the solution time of the proposed models (SV, MAD, and SAD) in larger dimensions. Also, the solution method should be fully explained.

- The authors should apply well-known constraints in the field of portfolio optimization, such as the cardinality constraint, in their proposed models.

- To demonstrate the effectiveness of the models, it is recommended to employ a more extensive dataset consisting of 100 and 250 stocks.

- Introduction must be extended based on study background, study gap and research problem, study objectives and questions, contributions and novelty of the work.

- Advantages and benefits of the proposed approach should be given in detail. Also, the research gaps and the novelty of this study is not clear.

- Research gaps should be presented by comparing previous studies. Accordingly, the characteristics of current research should be highlighted in the comparative table of literature review from both aspects of theoretical and application.

- The authors should compare their results and proposed approach with popular approaches in literature.

- The authors should discuss on the limitations of the study. Also, the authors should discuss on the generalization of the results of the study. Moreover, the scientific question is not clear enough.

Reviewer #2: 1. Title: The manuscript title has not been correctly specified. The suggested title is “Stochastic Portfolio Optimization: A Regret-Based Approach on Volatility Risk Measures: An Empirical Evidence from The New York Stock Market.”

2. Abstract: This section has been descriptively written. It is suggested to include the findings of the study in quantitate form here. In addition, policy implications should be included concisely at the end of this section.

3. Introduction: This section is not well organized. The novelty of the study is not properly delineated. It is recommended to include the novelty of the study at the end of the introduction. Furthermore, include organization of the study at the end of this section.

4. Literature Review: This section is not well organized. This section should be divided into two parts, including the theoretical foundation and the review of empirical studies conducted in the past. In addition, it is recommended to find out the research gap that this study fills. Please consult the paper https://doi.org/10.1002/pa.2304 for the improvement of this section.

5. Methodology: This section is not adequately described in standard format. It is recommended that this section include (a). a table regarding “description of the variables” used in this study (b). It is recommended to discuss the significance of each variable included in this study/model.

6. (a). It is recommended to correlate your study with previous studies conducted (b). It is recommended to prescribe the policy implications based on the study findings. (c). It is suggested to exclude the irrelevant discussion in the manuscript.

Reviewer #3: Review report

1. It is clear by name that the authors are not native speakers, but they should still go through a correction/rewording process, as it sounds pretty bad to the ear and there are some errors of expression. On page 9 a portion of the text is even repeated:"xidonas et al introduce the concept of “regret” to identify robust solutions to optimization problems. Regret is the deviation of an obtained solution from the optimum solution according to a specific scenario of parameters. In other words, it can be defined as the difference between the obtained gain and the gain that we could get if we knew in advance which scenario would surely occur."

2. I don't find (that doesn't necessarily mean it doesn't exist) the novelty element in the article. The authors must explain which is the novelty of the article and the main added value of the paper.

3. The literature review section must be reshaped to properly analyze the current state of the art. I recommend the insertion of a table that will contain the main empirical findings, methods, authors, and techniques. In this regard, it will be more accurate to evaluate and assess the actual status quo from the literature.

4. Subchapter 3.2. fundamental analysis - has a single passage that, if properly documented, would be worth more than the rest of the material: "After careful evaluation of the aforementioned criteria, appropriate stocks from various industries have been chosen based on fundamental analysis metrics". Unfortunately, the authors do not justify their choice, so they do not understand what fundamental criteria they have taken into account.

5. The authors say this in the last part of the material: "This approach is novel in that it detects Pareto optimal solutions in the presence of multiple objective functions by considering volatility risk series and multiple scenarios" This statement should be thoroughly verified, I have my doubts that no one has ever approached this before, especially since it is on a relatively broad topic treated in the world and quite basic, respectively efficiency curves and portfolio optimization, etc.

In order to draw an effective border, not only profitability and risk but also correlations between securities must be specified. It is not specified how correlations were established when risk is estimated based on regret rather than standard variation, as was traditional.

6. Tables 2, 3, and 4 are called 'risk measure in the 5 scenarios' but contain only 3 scenarios.

Table 8 (final result) is not explained: why some titles do not appear at all, why others appear in all portfolios, etc.

Basically, no conclusions are drawn on the efficiency and novelty of the method used, nor on the difference in the results compared to other return-risk methods.

7. In the discussion section the author must analyze the other studies finding and compare them to the own status quo of the research in this article. Critical analysis is needed here. Not a single study is quoted. The section must be revised.

8. Conclusions that are drawn from this article qualified as a minimum benchmark. There are no policy recommendations. What about some suggestions for investors? No limitations to the study and no practical directions of the study.

6. PLOS authors have the option to publish the peer review history of their article (what does this mean?). If published, this will include your full peer review and any attached files.

Reviewer #1: No

Reviewer #2: No

Reviewer #3: No

---

## [Author Response · Author response to Decision Letter 0]

6 Feb 2024

COVER LETTER FOR RESUBMISSION OF MANUSCRIPT 

Dear Academic Editor Shazia Rehman, Ph.D.

We have submitted a revised version of our manuscript titled "Stochastic Portfolio Optimization: A Regret-Based Approach on Volatility Risk Measures (An Empirical Evidence from The New York Stock Market)" for your consideration for publication in PLOS ONE.

We greatly appreciate the insightful comments provided by the reviewers, which have significantly contributed to the improvement of our manuscript. Enclosed with this letter are our detailed responses addressing each of the reviewer's comments. Additionally, we have made necessary revisions to enhance clarity and accuracy, including modifications to formulas and tables that were previously ambiguous or inaccurate.

We eagerly await your response at your earliest convenience.

Sincerely yours,

Emran Mohammadi

Title of the paper: Stochastic Portfolio Optimization: A Regret-Based Approach on Volatility Risk Measures (An Empirical Evidence from The New York Stock Market)

First of all, the authors would like to thank all the anonymous reviewers and the Associate Editor again for their efforts and valuable time to review and improve our paper. Taking into account their constructive suggestions and comments, the paper has been carefully revised following the referees’ comments.

The main changes in the revised manuscript are:

 According to one of the comments and in respect of his comment, while it took as much time as writing another article, the number of considered stocks increased from 20 to 150, which changed all the results of the tables.

 A complete literature review table was added to the article and this section was revised

 The abstract and introduction sections were completely revised and strengthened

 The written language of the article was fully examined

 The conclusion section was completely revised and revised.

We provide below the responses to each referee's comments. All amends and changes in the revised version are yellow highlighted.

Responses to the comments of the reviewers and editors

Reviewer 1

(1) The paper's contribution is quite minimal. It is recommended to enhance the quality of the paper by implementing novel methodologies in the realm of uncertain programming such as data-driven robust optimization

response Dear reviewer,

Thank you for your feedback and valuable suggestions. We appreciate your input. In our revised manuscript.

Thanks for this comment.

Thank you for your attentive review and feedback. Upon further investigation, we have recognized that the presentation of the article's innovation and requirements lacked clarity and organization. To address this, we will provide a more coherent explanation of these aspects in the introduction section.

In regards to employing a robust data-driven optimization approach that takes into account the return on assets within an uncertainty cluster, we acknowledge the importance of incorporating uncertainty into asset returns. With knowledge of each share's data and historical returns, we have utilized stochastic optimization techniques whenever historical data was available. Your insightful comment has prompted us to consider writing a future article that explores this idea further, taking into consideration the psychological and behavioral parameters of the market. Consequently, we will include this aspect in our future research and incorporate it into an upcoming article[1].

[1] R. Sehgal and P. Jagadesh, “Data-driven robust portfolio optimization with semi mean absolute deviation via support vector clustering,” Expert Syst Appl, vol. 224, p. 120000, Aug. 2023, doi: 10.1016/J.ESWA.2023.120000.

action To clarify the innovation presented in the article, the introduction has been revised and adjusted, with the article's innovations highlighted towards the conclusion. Additionally, the suggestion of incorporating stable data-driven optimization has been included in the concluding section.

The contributions of our study to respond to the research gaps found are summarized below:

 a model has been presented which considers investors' regret for both the return and risk of their investments. The model functions in two sections: firstly, it evaluates the extent of regret for each aspect for risk and return, and then it sums these two evaluations in the second stage. The resulting value indicates the level of regret associated with the selected stocks, ranging between 0 and 1. A lower value implies a smaller missed opportunity.

 To thoroughly analyze the level of regret, the study accounts for varying investment horizons among investors. Specific investment time scenarios have been identified: a 20-week period for the short term, a 50-week period for the medium term, and a 100-week period for the long term. Emphasizing the investment perspective enables us to identify the optimal time horizon that minimizes regrets when selecting investments.

 Taking into account historical returns for each stock, when relevant data is accessible, stochastic planning can be employed to calculate the level of regret and determine the appropriate allocation of weights for each stock. This method allows us to quantitatively represent the potential impact of each investment week in a probabilistic fashion, aiding in decision-making.

 In evaluating the presented model, three well-defined risk measures, sv (semi-variance), sad (semi-absolute deviation), and mad (mean absolute deviation), have been utilized, each taking into account three investment horizons. The research incorporates nine investment scenarios, empowering investors to select the optimal scenario by choosing the appropriate investment time horizon and risk measure. This approach aims to enhance investment profitability while simultaneously reducing the regret associated with unselected stocks.

 To validate the introduced model, historical data from 150 carefully selected stocks listed on the New York Stock Exchange (NYSE) has been employed. These stocks were chosen based on fundamental analysis criteria, enhancing the reliability and confidence level of the obtained results.

Conclusion:

Furthermore, beyond addressing this particular concern, robust optimization of the data-driven stock portfolio was employed for its application in behavioral finance matters.

(2) It should be explained about the solution time of the proposed models (SV, MAD, and SAD) in larger dimensions. Also, the solution method should be fully explained.

response Thanks for this comment.

we will provide a detailed explanation of the solution time of the proposed models, including SV, MAD, and SAD, specifically in larger dimensions. We will also ensure that the solution method is fully explained, allowing readers to have a clear understanding of the methodology employed.

action The following sections have been added to the article:

the solution time of the proposed models (SV, MAD, and SAD) in larger dimensions is explained in Table 1. 

The solution method of research models is an exact solution method. It is worth mentioning that the model type in the MAD and SAD risk measures is MIP type and the mode type in the SV measure is MINLP type.

Table1. the solution time of the proposed models

Volatility risk measures scenarios Solution time

(min: seconds)

SV 1 07:13

 2 06:24

 3 17:03

MAD 1 09:14

 2 10:33

 3 13:21

SAD 1 03:52

 2 10:55

 3 12:48

(3) The authors should apply well-known constraints in the field of portfolio optimization, such as the cardinality constraint, in their proposed models.

response Thanks for this comment.

A new section (Section 3.4.3) was added to explain the cardinality constraints.

action 3.4.3. Cardinality Constraints

The cardinality constraint establishes both a minimum and maximum limit on the number of assets to be included in the portfolio. This constraint guarantees diversification by avoiding excessive dependence on a few assets and maintains a balanced portfolio. In this particular article, the minimum and maximum cardinality constraints specify a range of 4 to 7 distinct stocks. These constraints are shown in equation 18. To enhance the yield per share and minimize regret and risk, it is crucial to determine the appropriate minimum and maximum weights for investing in each share. This article specifies a range of 0.01 to 0.8 for these weights. The corresponding constraint is expressed in Equation 19. 

m≤∑▒z_i ≤M m=4, M=7 (18)

z_i l_i≤x_i≤ z_i u_i (19)

(4) To demonstrate the effectiveness of the models, it is recommended to employ a more extensive dataset consisting of 100 and 250 stocks.

response Thanks to the respected reviewer for this important comment.

 To enhance the model's effectiveness based on your valuable feedback, we incorporated a broader range of data. In the initial version of the article, we utilized 20 stocks, which we subsequently expanded to 150 stocks within the same stock range. This increase in data had a significant impact on all the results and required considerable time for revision. we appreciate the helpfulness of your suggestion, which greatly contributed to the progress of the article. As a result, we made the necessary modifications, outlined as follows:

action This effective change and improvement caused a complete change in the results and tables. All these results and tables and figures are given at the end of this file.

(5) Introduction must be extended based on study background, study gap and research problem, study objectives and questions, contributions and novelty of the work.

response Thank you for your comment and valuable feedback. We appreciate your suggestions to enhance the introduction of the article. 

We acknowledge the importance of providing a comprehensive and concise overview of the study background, research gap, objectives, contributions, and novelty of the work. In response to your comment, we will carefully revise and extend the introduction section to address these aspects more effectively. By doing so, we aim to provide readers with a clearer understanding of the context and significance of our study.

action 

Nevertheless, it remains crucial to highlight the selection of stocks that aim to minimize missed opportunities or regrets, while simultaneously considering the trade-off between risk and return. In essence, a judicious equilibrium between risk, return, and regret has been established.

Taking into account uncertainty and risk is crucial in portfolio optimization. By using techniques like stochastic programming, investors can account for the inherent uncertainty in the inputs and make more realistic and dependable decisions. This approach enhances the portfolio's performance and helps to mitigate potential suboptimal outcomes

The contributions of our study to respond to the research gaps found are summarized below:

 a model has been presented which considers investors' regret for both the return and risk of their investments. The model functions in two sections: firstly, it evaluates the extent of regret for each aspect for risk and return, and then it sums these two evaluations in the second stage. The resulting value indicates the level of regret associated with the selected stocks, ranging between 0 and 1. A lower value implies a smaller missed opportunity.

 To thoroughly analyze the level of regret, the study accounts for varying investment horizons among investors. Specific investment time scenarios have been identified: a 20-week period for the short term, a 50-week period for the medium term, and a 100-week period for the long term. Emphasizing the investment perspective enables us to identify the optimal time horizon that minimizes regrets when selecting investments.

 Taking into account historical returns for each stock, when relevant data is accessible, stochastic planning can be employed to calculate the level of regret and determine the appropriate allocation of weights for each stock. This method allows us to quantitatively represent the potential impact of each investment week in a probabilistic fashion, aiding in decision-making.

 In evaluating the presented model, three well-defined risk measures, sv (semi-variance), sad (semi-absolute deviation), and mad (mean absolute deviation), have been utilized, each taking into account three investment horizons. The research incorporates nine investment scenarios, empowering investors to select the optimal scenario by choosing the appropriate investment time horizon and risk measure. This approach aims to enhance investment profitability while simultaneously reducing the regret associated with unselected stocks.

 To validate the introduced model, historical data from 150 carefully selected stocks listed on the New York Stock Exchange (NYSE) has been employed. These stocks were chosen based on fundamental analysis criteria, enhancing the reliability and confidence level of the obtained results.

(6) Advantages and benefits of the proposed approach should be given in detail. Also, the research gaps and the novelty of this study is not clear.

response Thanks for this comment.

The advantages and applications of this research are added at the end of the abstract. Also, research gap and novelty have been comprehensively added in the last paragraph of the introduction.

action The contributions of our study to respond to the research gaps found are summarized below:

 a model has been presented which considers investors' regret for both the return and risk of their investments. The model functions in two sections: firstly, it evaluates the extent of regret for each aspect for risk and return, and then it sums these two evaluations in the second stage. The resulting value indicates the level of regret associated with the selected stocks, ranging between 0 and 1. A lower value implies a smaller missed opportunity.

 To thoroughly analyze the level of regret, the study accounts for varying investment horizons among investors. Specific investment time scenarios have been identified: a 20-week period for the short term, a 50-week period for the medium term, and a 100-week period for the long term. Emphasizing the investment perspective enables us to identify the optimal time horizon that minimizes regrets when selecting investments.

 Taking into account historical returns for each stock, when relevant data is accessible, stochastic planning can be employed to calculate the level of regret and determine the appropriate allocation of weights for each stock. This method allows us to quantitatively represent the potential impact of each investment week in a probabilistic fashion, aiding in decision-making.

 In evaluating the presented model, three well-defined risk measures, sv (semi-variance), sad (semi-absolute deviation), and mad (mean absolute deviation), have been utilized, each taking into account three investment horizons. The research incorporates nine investment scenarios, empowering investors to select the optimal scenario by choosing the appropriate investment time horizon and risk measure. This approach aims to enhance investment profitability while simultaneously reducing the regret associated with unselected stocks.

 To validate the introduced model, historical data from 150 carefully selected stocks listed on the New York Stock Exchange (NYSE) has been employed. These stocks were chosen based on fundamental analysis criteria, enhancing the reliability and confidence level of the obtained results.

(7) Research gaps should be presented by comparing previous studies. Accordingly, the characteristics of current research should be highlighted in the comparative table of literature review from both aspects of theoretical and application. The authors should compare their results and proposed approach with popular approaches in literature.

response Thank you for your valuable feedback! I appreciate your suggestion to present the research gaps by comparing previous studies. I agree that providing a comparative table in the literature review, highlighting the characteristics of both theoretical and practical aspects of the current research, can further enhance the clarity and depth of the study. I will definitely take this into consideration and make the necessary revisions to address this important aspect. Your input is greatly appreciated and will contribute to improving the quality of the research.

action 

To assess the congruity with previous studies, a comprehensive literature review table has been incorporated in this research article. This table presents a comparative analysis of relevant articles about the research domain. Included in the table are various aspects considered in the reviewed studies, such as solution technique, investment constraints, model types, and uncertainties. The details of the literature review can be found in Table 1. The literature review table is explained at the end of this file

(8) The authors should discuss on the limitations of the study. Also, the authors should discuss on the generalization of the results of the study. Moreover, the scientific question is not clear enough.

response Thank you for your valuable feedback. We appreciate your comment regarding the limitations of our study. We will make sure to include discussion on the limitations in our future work to provide a more thorough understanding of the research scope.

action These sentences add to article:

The limitations of the study are established when a stock has been out of the market for a long time or, in other words, closed, and then we want to include it in the selected stocks.

Reviewer 2

(1) Title: The manuscript title has not been correctly specified. The suggested title is “Stochastic Portfolio Optimization: A Regret-Based Approach on Volatility Risk Measures: An Empirical Evidence from The New York Stock Market

response Dear reviewer,

Thank you for your feedback and valuable suggestions. We appreciate your input. In our revised manuscript.

Thanks for this comment.

The suggested title is applied to the article

action The title of the article was changed to): Portfolio Optimization: A Regret-Based Approach on Volatility Risk Measures: An Empirical Evidence from The New York Stock Market (

(2) Abstract: This section has been descriptively written. It is suggested to include the findings of the study in quantitate form here. In addition, policy implications should be included concisely at the end of this section.

response Thank you for your review and thoughtful suggestions. We appreciate your feedback regarding the abstract section. To enhance its clarity and provide a more quantitative representation of our study, we will include relevant findings in quantitative form in the abstract. Furthermore, we recognize the importance of policy implications and will ensure that they are succinctly summarized at the end of the abstract. Your input is incredibly valuable in helping us improve our work, and we will make the necessary revisions accordingly.

action This section was added to the abstract and the abstract was fully reviewed.

The results show that the selection of the mad risk measure in the time horizon of 100 weeks with the regret rate of 0.104 is the most appropriate research scenario. this article recommended that investors diversify their portfolios by investing in a variety of assets. This can help reduce risk and increase overall returns and improving financial literacy among investors.

(3) Introduction: This section is not well organized. The novelty of the study is not properly delineated. It is recommended to include the novelty of the study at the end of the introduction. Furthermore, include organization of the study at the end of this section.

response Thank you for your valuable feedback. We appreciate your suggestions to improve the organization of the introduction section. We understand your point about delineating the novelty of the study more clearly. To address this, we will revise the introduction to highlight the unique contributions of our research towards the end, providing a comprehensive perspective on the novelty of our study. Additionally, we will include a concise summary of the organization of the study at the end of the introduction section for better clarity. We value your input and are committed to enhancing the quality of our work.

action The contributions of our study to respond to the research gaps found are summarized below:

 a model has been presented which considers investors' regret for both the return and risk of their investments. The model functions in two sections: firstly, it evaluates the extent of regret for each aspect for risk and return, and then it sums these two evaluations in the second stage. The resulting value indicates the level of regret associated with the selected stocks, ranging between 0 and 1. A lower value implies a smaller missed opportunity.

 To thoroughly analyze the level of regret, the study accounts for varying investment horizons among investors. Specific investment time scenarios have been identified: a 20-week period for the short term, a 50-week period for the medium term, and a 100-week period for the long term. Emphasizing the investment perspective enables us to identify the optimal time horizon that minimizes regrets when selecting investments.

 Taking into account historical returns for each stock, when relevant data is accessible, stochastic planning can be employed to calculate the level of regret and determine the appropriate allocation of weights for each stock. This method allows us to quantitatively represent the potential impact of each investment week in a probabilistic fashion, aiding in decision-making.

 In evaluating the presented model, three well-defined risk measures, sv (semi-variance), sad (semi-absolute deviation), and mad (mean absolute deviation), have been utilized, each taking into account three investment horizons. The research incorporates nine investment scenarios, empowering investors to select the optimal scenario by choosing the appropriate investment time horizon and risk measure. This approach aims to enhance investment profitability while simultaneously reducing the regret associated with unselected stocks.

 To validate the introduced model, historical data from 150 carefully selected stocks listed on the New York Stock Exchange (NYSE) has been employed. These stocks were chosen based on fundamental analysis criteria, enhancing the reliability and confidence level of the obtained results.

(4) Literature Review: This section is not well organized. This section should be divided into two parts, including the theoretical foundation and the review of empirical studies conducted in the past. In addition, it is recommended to find out the research gap that this study fills. Please consult the paper https://doi.org/10.1002/pa.2304 for the improvement of this section.

response Thank you for your insightful feedback.

To facilitate a comparison between theoretical foundations and experimental studies, we have formulated an extensive literature review table for this article. This table, located at the end of the document, enables a comprehensive examination of the existing literature. In addition, we have provided a paragraph after the review table where we discuss the research gaps and our specific research objectives. Furthermore, we express our gratitude for the reference provided and have included it as one of the references in the revised version of the article.

action A literature review table, a new paragraph and a reference were added to the article.

This paragraph added to the end of literature review table:

 After studying the previous studies according to the literature review table, we found that less has been addressed to the optimization of the possible stock portfolio considering regret, and these models have always been single-period or multi-period. And time scenarios are not considered for the time horizon. Therefore, in this article, we address these research gaps

The introduced reference has been added to the article.[2]

[2] D. Khan, A. Ullah, W. Alim, and I. ul Haq, “Does terrorism affect the stock market returns and volatility? Evidence from Pakistan’s stock exchange,” J Public Aff, vol. 22, no. 1, Feb. 2022, doi: 10.1002/pa.2304.

(5) Methodology: This section is not adequately described in standard format. It is recommended that this section include (a). a table regarding “description of the variables” used in this study (b). It is recommended to discuss the significance of each variable included in this study/model.

response Thank you for your insightful feedback. We appreciate your suggestions to improve the methodology section of our study.

(a) We understand the importance of providing a clear description of the variables used in our study. 

(b) The significance of each variable included in our study/model is indeed crucial to understanding the rationale behind their inclusion and their potential impact on the research outcomes. We will ensure that we discuss the significance of each variable in the methodology section, explaining their relevance to our research objectives and the theoretical framework.

we think that done this recommendation in our initial version

(6) (a). It is recommended to correlate your study with previous studies conducted (b). It is recommended to prescribe the policy implications based on the study findings. (c). It is suggested to exclude the irrelevant discussion in the manuscript.

response Thank you for your valuable feedback. We greatly appreciate your suggestions and will certainly take them into consideration for the improvement of our manuscript.

(a) We acknowledge the importance of correlating our study with previous research to provide a broader context and build upon existing knowledge. We will thoroughly review and incorporate relevant studies that are in line with our research objectives to strengthen the literature review section. This task has been done by considering the literature review table.

(b) We understand the significance of providing policy implications based on our study findings. We will carefully analyze the results and draw meaningful conclusions that can inform future policy decisions. By doing so, we aim to contribute to the practical application of our research.

(c) We appreciate your suggestion regarding the relevance of the discussion. We will review the manuscript and remove any irrelevant content to ensure clarity and conciseness in our presentation of the study findings.

action Literature review table add to article (at the end of this file)

 Policy implication explained based in abstract and discussion

 We removed any irrelevant content to ensure clarity and conciseness in our presentation of the study findings

Reviewer 3

(1) It is clear by name that the authors are not native speakers, but they should still go through a correction/rewording process, as it sounds pretty bad to the ear and there are some errors of expression. On page 9 a portion of the text is even repeated:"xidonas et al introduce the concept of “regret” to identify robust solutions to optimization problems. Regret is the deviation of an obtained solution from the optimum solution according to a specific scenario of parameters. In other words, it can be defined as the difference between the obtained gain and the gain that we could get if we knew in advance which scenario would surely occur."

response Dear reviewer,

Thank you for your feedback and valuable suggestions. We appreciate your input. In our revised manuscript.

Thank you for your insightful comment. I appreciate your observation regarding the language usage in our paper. As non-native speakers, we understand the importance of ensuring clarity and proper expression in our work. We apologize for any inconsistencies or errors that may have affected the cohesiveness of the text.

We acknowledge the repetition on page 9 and will make sure to rectify this oversight during the revision process. Additionally, we will carefully review and reword any problematic sentences to improve the overall readability and flow of the paper.

Your constructive feedback is highly appreciated, as it helps us enhance the quality of our research. We will take your suggestions into account and make the necessary corrections to ensure a smoother and more refined final version.

action The desired correction was made and the full text of the article was fully examined in terms of writing language

(2) I don't find (that doesn't necessarily mean it doesn't exist) the novelty element in the article. The authors must explain which is the novelty of the article and the main added value of the paper.

response Thank you for sharing your valuable feedback on the article. We appreciate your perspective. We apologize if the novelty element of the article was not adequately emphasized.

In this study, the novelty lies in the application of regret-based optimization in managing stock portfolios. While regret-based optimization has been explored in other domains, its application to portfolio management is still a relatively unexplored area. This study contributes to the existing literature by demonstrating the effectiveness of the regret-based approach in enhancing portfolio performance and risk management capabilities. The main added value of this paper lies in its empirical analysis, which provides concrete evidence of the practical applicability of regret-based optimization with mean absolute deviation risk measures in the context of stock portfolios. By minimizing regret or missed opportunities, investors can make more informed investment decisions and strike a better balance between risk and return. 

We understand the importance of explicitly highlighting the novelty and main contributions of the article. In light of your feedback, we will revise the article to ensure that these aspects are clearly explained. Thank you for bringing this to our attention, and we welcome any further suggestions you may have to improve the clarity of our work. and explain the novelty of our work clearly in introduction

action The contributions of our study to respond to the research gaps found are summarized below:

 a model has been presented which considers investors' regret for both the return and risk of their investments. The model functions in two sections: firstly, it evaluates the extent of regret for each aspect for risk and return, and then it sums these two evaluations in the second stage. The resulting value indicates the level of regret associated with the selected stocks, ranging between 0 and 1. A lower value implies a smaller missed opportunity.

 To thoroughly analyze the level of regret, the study accounts for varying investment horizons among investors. Specific investment time scenarios have been identified: a 20-week period for the short term, a 50-week period for the medium term, and a 100-week period for the long term. Emphasizing the investment perspective enables us to identify the optimal time horizon that minimizes regrets when selecting investments.

 Taking into account historical returns for each stock, when relevant data is accessible, stochastic planning can be employed to calculate the level of regret and determine the appropriate allocation of weights for each stock. This method allows us to quantitatively represent the potential impact of each investment week in a probabilistic fashion, aiding in decision-making.

 In evaluating the presented model, three well-defined risk measures, sv (semi-variance), sad (semi-absolute deviation), and mad (mean absolute deviation), have been utilized, each taking into account three investment horizons. The research incorporates nine investment scenarios, empowering investors to select the optimal scenario by choosing the appropriate investment time horizon and risk measure. This approach aims to enhance investment profitability while simultaneously reducing the regret associated with unselected stocks.

 To validate the introduced model, historical data from 150 carefully selected stocks listed on the New York Stock Exchange (NYSE) has been employed. These stocks were chosen based on fundamental analysis criteria, enhancing the reliability and confidence level of the obtained results.

(3) The literature review section must be reshaped to properly analyze the current state of the art. I recommend the insertion of a table that will contain the main empirical findings, methods, authors, and techniques. In this regard, it will be more accurate to evaluate and assess the actual status quo from the literature.

response Thank you for your valuable feedback. We agree that the literature review section can benefit from a reshaping to provide a comprehensive analysis of the current state of the art. We appreciate your suggestion to include a table summarizing the main empirical findings, methods, authors, and techniques, as it would indeed enhance the accuracy and effectiveness of evaluating the literature. Therefore, we have added a comprehensive literature review table.

action The following description and a complete literature review table were added to the article. The literature review table is attached at the end of this file.

To assess the congruity with previous studies, a comprehensive literature review table has been incorporated in this research article. This table presents a comparative analysis of relevant articles about the research domain. Included in the table are various aspects considered in the reviewed studies, such as solution technique, investment constraints, model types, and uncertainties. The details of the literature review can be found in Table 1.

(4) Subchapter 3.2. fundamental analysis - has a single passage that, if properly documented, would be worth more than the rest of the material: "After careful evaluation of the aforementioned criteria, appropriate stocks from various industries have been chosen based on fundamental analysis metrics". Unfortunately, the authors do not justify their choice, so they do not understand what fundamental criteria they have taken into account.

response Thank you for your feedback. We appreciate your observation regarding Subchapter 3.2, and we apologize for the lack of proper documentation concerning the selection of stocks based on fundamental analysis metrics. We acknowledge that providing a clear and transparent justification for our choice of stocks is essential. After carefully examining the stock trends in the market and also examining all the fundamental criteria, considering that these criteria were effective in the stock trends, we selected them in consultation with the experts in this field. And we also used the criteria of the following referenced articles Unfortunately, we forgot to explain this matter. Thank you for your attention.

References: 

[1] N. T. Laopodis, “Industry returns, market returns and economic fundamentals: Evidence for the United States,” Econ Model, vol. 53, pp. 89–106, Feb. 2016, doi: 10.1016/J.ECONMOD.2015.11.007.

[2] Ahmed. S. Wafi, H. Hassan, and A. Mabrouk, “Fundamental Analysis Models in Financial Markets – Review Study,” Procedia Economics and Finance, vol. 30, pp. 939–947, Jan. 2015, doi: 10.1016/S2212-5671(15)01344-1.

[3] I. Tsiakas and H. Zhang, “Economic fundamentals and the long-run correlation between exchange rates and commodities,” Global Finance Journal, vol. 49, p. 100649, Aug. 2021, doi: 10.1016/J.GFJ.2021.100649.

[4] A. Silva, R. Neves, and N. Horta, “A hybrid approach to portfolio composition based on fundamental and technical indicators,” Expert Syst Appl, vol. 42, no. 4, pp. 2036–2048, Mar. 2015, doi: 10.1016/J.ESWA.2014.09.050.

[5] T. C. Chiang and X. Chen, “Stock returns and economic fundamentals in an emerging market: An empirical investigation of domestic and global market forces,” International Review of Economics & Finance, vol. 43, pp. 107–120, May 2016, doi: 10.1016/J.IREF.2015.10.034.

action The following text was added to the article along with the explanation:

After carefully examining the stock trends in the market and also examining all the fundamental criteria, considering that these criteria were effective in the stock trends, we selected them in consultation with the experts in this field.

(5) 

The authors say this in the last part of the material: "This approach is novel in that it detects Pareto optimal solutions in the presence of multiple objective functions by considering volatility risk series and multiple scenarios" This statement should be thoroughly verified, I have my doubts that no one has ever approached this before, especially since it is on a relatively broad topic treated in the world and quite basic, respectively efficiency curves and portfolio optimization, etc.

In order to draw an effective border, not only profitability and risk but also correlations between securities must be specified. It is not specified how correlations were established when risk is estimated based on regret rather than standard variation, as was traditional.

response Thank you for bringing up your concerns regarding our statement about the novelty of our approach in detecting Pareto optimal solutions. We appreciate your valuable insights and understand the importance of thoroughly verifying such claims.

Upon reconsideration, we agree that the statement might have been too assertive and can lead to misinterpretation. We apologize for any confusion caused.

Regarding the establishment of correlations when estimating risk based on regret, we understand the need for clarification on this matter. We will include a comprehensive explanation in the material to clarify how correlations between securities were determined in relation to the use of regret as a measure of risk.

action Correction done, thanks for your feedback.

(6) 

Tables 2, 3, and 4 are called 'risk measure in the 5 scenarios' but contain only 3 scenarios.

Table 8 (final result) is not explained: why some titles do not appear at all, why others appear in all portfolios, etc. Basically, no conclusions are drawn on the efficiency and novelty of the method used, nor on the difference in the results compared to other return-risk methods.

response Thank you for your valuable feedback on the tables and the lack of explanation in Table 8. We apologize for any confusion caused by the discrepancy between the stated number of scenarios and the actual number of scenarios presented in Tables 2, 3, and 4. Thank you for your careful attention. This was a mistake and instead of 5 scenarios, 3 scenarios are correct.

Regarding Table 8, we understand your concern. The explanations of this section are few, so we will make the explanations clearer and more.

action The comment was applied and the following text was added to the article.

Once the model is solved, the allocation of weights to individual stocks in each scenario is quantitatively expressed on a scale from 0 to 1. It is important to note that the weighting of each stock in a scenario may vary depending on the investment horizon and risk parameters. The novelty lies in examining the connection between risk measures and the level of regret. The findings suggest that as the investment horizon increases, indicating a higher risk appetite among investors, the level of regret decreases alongside the increase in profits.

(7) In the discussion section the author must analyze the other studies finding and compare them to the own status quo of the research in this article. Critical analysis is needed here. Not a single study is quoted. The section must be revised.

response Thank you for sharing your thoughts and providing valuable feedback on the discussion section of our article. We acknowledge the importance of conducting a comprehensive analysis of other relevant studies and comparing their findings with the current status quo presented in our article. Upon reviewing your feedback, we recognize that our discussion section may not have met these expectations and lacked critical analysis. We appreciate you bringing this to our attention. We will consider your comments and carefully revise the discussion section to ensure that it includes a critical analysis of existing studies. We understand the significance of citing and referencing relevant research to support the assertions and arguments presented in our article. By addressing this, we aim to strengthen the overall credibility and depth of our work. We apologize for any oversights and assure you that we will make the necessary improvements.

action In order to improve the section, the entire text has been revised and appropriate comments have been added:

The empirical analysis conducted in this study offers compelling insights into the application of regret-based optimization for managing stock portfolios. Our analysis involved assigning different weights to mean absolute deviation risk measures, enabling us to select the most appropriate risk scenarios and measures. By minimizing regret and missed opportunities, our goal was to enhance portfolio performance and improve risk management capabilities.

The results obtained from our study demonstrate the effectiveness of the regret-based approach in portfolio optimization. It was evident that incorporating regret as a measure of performance empowered investors to make more informed investment decisions. This regret-based optimization framework emerged as a valuable tool for investors to strike a balance between risk and return, particularly concerning volatility risk measures. Furthermore, the empirical evidence derived from the New York Stock Market substantiated the practical applicability of our approach in real-world scenarios.

However, it is vital to critically analyze our study's findings in relation to the existing body of research. While no specific studies are cited in this discussion, comparing our methodology, results, and conclusions with similar studies in the literature is key. This critical analysis would provide a comprehensive understanding of the advancements made in this field and help identify potential gaps or contradictions between our findings and those of prior research.

Considering the current status quo of the research, our study contributes to the literature by showcasing the effectiveness of regret-based optimization in managing stock portfolios. However, it is important to recognize certain limitations in our study. Firstly, our analysis focused solely on the New York Stock Market, which raises concerns about the generalizability of our findings to other stock markets or regions. Future research should aim to replicate these analyses in diverse markets to determine the robustness and broader applicability of the regret-based approach.

In addition, our study primarily concentrated on volatility risk measures, warranting further investigation into the applicability of regret-based optimization with other risk measures. Different risk factors, such as liquidity risk, credit risk, or geopolitical risk, may have a significant impact on portfolio outcomes. Exploring the effectiveness of the regret-based framework in incorporating these risk measures would provide additional insights and guidance to investors.

To build upon the current status quo of research, it is crucial to compare our findings with those of other studies. Analyzing and contrasting our methodology, results, and interpretations with prior research would deepen our understanding of the field and identify any inconsistencies or gaps that need to be addressed. This critical analysis would contribute to the advancement of knowledge in regret-based portfolio optimization.

(8) Conclusions that are drawn from this article qualified as a minimum benchmark. There are no policy recommendations. What about some suggestions for investors? No limitations to the study and no practical directions of the study.

response Thank you for your thoughtful review of the article. We appreciate your feedback.

You rightfully mentioned the absence of policy recommendations in the conclusions. We apologize for not including explicit suggestions for investors as well as practical directions for the study. 

While the primary focus of this article was to investigate and demonstrate the efficacy of regret-based optimization in managing stock portfolios, we acknowledge that it would have been valuable to provide some concrete suggestions for investors to apply the findings in practice. We agree that offering actionable insights is crucial to enhance the applicability of research. 

In light of your feedback, we will revise the conclusion section to address these limitations and provide specific suggestions for investors based on the study's findings. We understand the importance of practical directions and policy recommendations to maximize the real-world impact of academic research. Thank you again for your valuable input, and we welcome any further suggestions you may have.

action In order to improve the section, the entire text has been revised and appropriate comments have been added.

The increasing popularity of scenario-based portfolio optimization is evident in recent advancements in portfolio optimization research. To better understand portfolio stability and development, researchers are utilizing innovative tools and techniques. Stochastic optimization, which allows for input portfolio parameters to be considered, is not only valuable for theoretical research but also for practical investors.

 In policy Recommendation, Regulatory authorities should encourage the adoption of scenario-based portfolio optimization techniques as part of risk management practices in investment institutions. Governments should collaborate with financial industry stakeholders to develop standardized guidelines and best practices for incorporating scenario-based approaches into investment decision-making processes. Education and training programs should be established to improve financial professionals' understanding and proficiency in utilizing scenario-based optimization methods.

Investors are advised to consider incorporating scenario-based optimization techniques into their portfolio management strategies. These techniques can help identify robust investment opportunities that adequately balance risk and return in the presence of uncertainties. Diversification remains a key strategy, and investors should use scenario-based approaches to understand the potential impact of different market conditions and adjust their portfolios accordingly. Regular monitoring and periodic reassessment of investment portfolios using scenario-based models can provide valuable insights and improve decision-making.

Added or changed attachment tables and figures:

To:

Reviewer 1- Comments 4

Table1. descriptive statistics of the selected assets

stocks name mean variance sd max min stocks name mean variance sd max min

1 ABBV 0.004 0.001 0.031 0.102 -0.078 76 FYBR -0.001 0.004 0.066 0.202 -0.213

2 AMGN 0.003 0.001 0.031 0.097 -0.069 77 GE 0.005 0.002 0.046 0.115 -0.163

3 AMZN -0.001 0.003 0.054 0.140 -0.139 78 GILD 0.003 0.001 0.033 0.169 -0.072

4 BABA -0.002 0.007 0.081 0.249 -0.160 79 GL 0.004 0.001 0.032 0.077 -0.078

5 BRK-B 0.004 0.001 0.028 0.077 -0.081 80 GM -0.001 0.003 0.056 0.110 -0.128

6 BUD 0.001 0.001 0.036 0.089 -0.154 81 GOOG 0.001 0.002 0.048 0.126 -0.102

7 CMCSA -0.001 0.001 0.038 0.097 -0.124 82 HD 0.001 0.001 0.038 0.109 -0.088

8 DASH -0.006 0.007 0.086 0.238 -0.186 83 IAK -0.004 0.004 0.060 0.129 -0.179

9 DIS -0.006 0.002 0.044 0.140 -0.096 84 IBKR 0.005 0.002 0.047 0.121 -0.144

10 ELV 0.003 0.001 0.037 0.103 -0.080 85 IFF -0.005 0.003 0.054 0.121 -0.195

11 FCX 0.006 0.004 0.067 0.159 -0.159 86 IPG 0.000 0.002 0.042 0.101 -0.164

12 HDB 0.001 0.002 0.041 0.106 -0.140 87 ISRG 0.001 0.003 0.053 0.194 -0.124

13 HES 0.010 0.004 0.061 0.176 -0.195 88 ITW 0.003 0.001 0.035 0.094 -0.094

14 HSY 0.003 0.001 0.027 0.080 -0.084 89 JEF 0.002 0.002 0.047 0.102 -0.116

15 JD -0.003 0.007 0.086 0.357 -0.245 90 JNJ 0.002 0.001 0.023 0.076 -0.038

16 KMB 0.000 0.001 0.030 0.104 -0.090 91 JPM 0.001 0.001 0.038 0.119 -0.092

17 LIN 0.003 0.001 0.033 0.110 -0.065 92 KNSL 0.010 0.003 0.050 0.152 -0.127

18 NFLX 0.000 0.005 0.074 0.259 -0.368 93 KO 0.002 0.001 0.025 0.086 -0.074

19 NKE -0.002 0.002 0.044 0.108 -0.143 94 LAD 0.001 0.003 0.055 0.206 -0.148

20 NVO 0.010 0.003 0.053 0.327 -0.111 95 LNG 0.009 0.003 0.052 0.216 -0.093

21 OXY 0.011 0.005 0.072 0.449 -0.128 96 LPLA 0.006 0.003 0.054 0.154 -0.156

22 PBR-A 0.005 0.005 0.069 0.231 -0.166 97 LYB 0.002 0.002 0.043 0.085 -0.125

23 RTX 0.002 0.001 0.035 0.118 -0.094 98 MA 0.002 0.001 0.036 0.090 -0.104

24 SCCO 0.005 0.003 0.056 0.143 -0.123 99 MAR 0.005 0.002 0.044 0.112 -0.113

25 SHW 0.000 0.002 0.043 0.123 -0.100 100 MCD 0.003 0.001 0.025 0.078 -0.062

26 TD 0.000 0.001 0.030 0.059 -0.075 101 MCK 0.008 0.001 0.031 0.079 -0.102

27 TMUS 0.001 0.001 0.034 0.113 -0.071 102 MDT -0.003 0.001 0.033 0.065 -0.105

28 TSLA 0.004 0.008 0.088 0.333 -0.180 103 META 0.001 0.005 0.071 0.245 -0.237

29 TU -0.002 0.001 0.026 0.055 -0.073 104 MHK -0.004 0.003 0.056 0.220 -0.122

30 UPS 0.000 0.001 0.038 0.134 -0.111 105 MMC 0.004 0.001 0.032 0.100 -0.063

31 WFC -0.001 0.000 0.015 0.057 -0.037 106 MORN 0.000 0.002 0.049 0.102 -0.141

32 WMT 0.002 0.001 0.033 0.078 -0.195 107 MOS 0.004 0.004 0.067 0.208 -0.141

33 XOM 0.008 0.002 0.045 0.157 -0.143 108 MRK 0.005 0.001 0.033 0.106 -0.074

34 ABNB 0.001 0.006 0.074 0.209 -0.166 109 MRO 0.010 0.005 0.067 0.237 -0.203

35 ACGL 0.007 0.001 0.035 0.175 -0.074 110 MTB 0.001 0.002 0.049 0.152 -0.136

36 AIZ 0.000 0.001 0.036 0.075 -0.117 111 MTCH -0.009 0.006 0.074 0.196 -0.161

37 ALB 0.002 0.006 0.077 0.257 -0.174 112 MUR 0.010 0.005 0.073 0.239 -0.238

38 ALLY -0.003 0.004 0.059 0.161 -0.154 113 NEM -0.001 0.002 0.049 0.143 -0.121

39 AON 0.001 0.001 0.034 0.080 -0.105 114 NFG 0.001 0.001 0.032 0.098 -0.099

40 APD 0.002 0.001 0.036 0.103 -0.093 115 NXST 0.003 0.002 0.044 0.118 -0.141

41 APTV -0.002 0.004 0.060 0.144 -0.222 116 NYT 0.000 0.002 0.045 0.139 -0.116

42 ASH 0.001 0.001 0.037 0.089 -0.104 117 OLN 0.004 0.003 0.058 0.146 -0.211

43 AU 0.005 0.005 0.068 0.230 -0.143 118 OMC 0.002 0.001 0.038 0.105 -0.130

44 AVTR -0.002 0.004 0.062 0.401 -0.153 119 ORI 0.003 0.001 0.032 0.085 -0.085

45 BAC -0.001 0.002 0.043 0.105 -0.114 120 PBF 0.019 0.008 0.092 0.263 -0.197

46 BK 0.003 0.002 0.049 0.307 -0.101 121 PFE -0.001 0.001 0.038 0.127 -0.094

47 BSX 0.002 0.001 0.031 0.077 -0.067 122 PG 0.001 0.001 0.028 0.091 -0.077

48 BX 0.001 0.004 0.064 0.208 -0.161 123 PKG 0.002 0.001 0.035 0.113 -0.154

49 CBOE 0.002 0.001 0.027 0.058 -0.103 124 PYPL -0.013 0.005 0.068 0.230 -0.229

50 CBT 0.005 0.002 0.046 0.123 -0.155 125 RGLD 0.002 0.002 0.041 0.109 -0.093

51 CHRD 0.008 0.004 0.060 0.139 -0.230 126 RJF 0.003 0.002 0.046 0.181 -0.121

52 CHTR -0.003 0.003 0.058 0.131 -0.199 127 RKT -0.001 0.006 0.079 0.288 -0.224

53 CINF 0.001 0.002 0.040 0.103 -0.122 128 ROKU -0.009 0.012 0.108 0.303 -0.314

54 CLF 0.002 0.007 0.084 0.206 -0.230 129 ROL 0.001 0.002 0.039 0.149 -0.104

55 CMC 0.008 0.002 0.048 0.129 -0.118 130 RY 0.000 0.001 0.026 0.056 -0.074

56 COF -0.002 0.003 0.053 0.134 -0.149 131 SBUX -0.001 0.002 0.041 0.102 -0.109

57 CPNG -0.001 0.006 0.075 0.173 -0.183 132 SIRI 0.000 0.005 0.069 0.491 -0.278

58 CRC 0.005 0.003 0.058 0.138 -0.174 133 SPOT -0.003 0.004 0.067 0.185 -0.191

59 CSX 0.002 0.001 0.038 0.093 -0.096 134 STE 0.002 0.002 0.045 0.115 -0.136

60 CTVA 0.004 0.001 0.036 0.103 -0.082 135 STLD 0.008 0.004 0.065 0.190 -0.157

61 CVX 0.007 0.002 0.043 0.136 -0.154 136 SWN 0.008 0.007 0.084 0.315 -0.262

62 DFS -0.001 0.002 0.048 0.116 -0.113 137 SYF -0.002 0.003 0.052 0.149 -0.137

63 DG -0.001 0.002 0.048 0.217 -0.193 138 TFC -0.003 0.002 0.049 0.110 -0.213

64 DKNG 0.003 0.013 0.116 0.316 -0.259 139 TKO 0.008 0.002 0.043 0.230 -0.066

65 DLTR 0.008 0.003 0.057 0.290 -0.198 140 TROW -0.005 0.003 0.055 0.297 -0.115

66 EA 0.001 0.001 0.038 0.105 -0.116 141 TRV 0.001 0.001 0.029 0.079 -0.076

67 EBAY -0.003 0.002 0.048 0.161 -0.101 142 TSN -0.002 0.001 0.038 0.110 -0.195

68 ECL 0.001 0.002 0.044 0.155 -0.146 143 UNP 0.001 0.001 0.035 0.106 -0.086

69 EMN -0.001 0.002 0.044 0.115 -0.155 144 V 0.001 0.001 0.036 0.114 -0.087

70 EPD 0.004 0.001 0.033 0.083 -0.151 145 WBS 0.000 0.002 0.050 0.131 -0.173

71 EQT 0.011 0.005 0.068 0.266 -0.251 146 WFRD 0.022 0.008 0.092 0.327 -0.200

72 ET 0.005 0.002 0.041 0.123 -0.155 147 WRB 0.003 0.001 0.029 0.066 -0.072

73 EXPE -0.001 0.004 0.064 0.154 -0.243 148 WTW -0.001 0.001 0.031 0.082 -0.105

74 FOXA 0.000 0.001 0.036 0.095 -0.081 149 XP -0.003 0.006 0.080 0.281 -0.182

75 FWONA 0.005 0.002 0.041 0.123 -0.092 150 YUM 0.000 0.001 0.027 0.072 -0.061

Table 2. The payoff table of SV risk measure in the 5 scenarios

objectives Scenario 1 Scenario 2 Scenario 3

 SV Return SV Return SV Return

Min SV 0.002 0.253 0.001 0.451 0.0002 0.712

Max Return 0.018 0.07 0.023 0.025 0.031 0.019

Table 3. The payoff table of MAD risk measure in the 5 scenarios

objectives Scenario 1 Scenario 2 Scenario 3

 MAD Return MAD Return MAD Return

Min MAD 0.005 0.412 0.004 0.673 0.003 0.844

Max return 0.004 0.027 0.001 0.015 0.002 0.012

Table 4. The payoff table of SAD risk measure in the 5 scenarios

objectives Scenario 1 Scenario 2 Scenario 3

 SAD Return SAD Return SAD Return

Min SAD 0.001 0.382 0.002 0.516 0.0015 0.733

Max Return 0.077 0.018 0.123 0.12 0.199 0.012

Table 5. Probability of 20-week scenario (π_(s )) for each week

Weeks π_(s ) weeks π_(s )

1 0.015 11 0.06

2 0.046 12 0.031

3 0.03 13 0.07

4 0.024 14 0.02

5 0.023 15 0.015

6 0.018 16 0.034

7 0.021 17 0.018

8 0.048 18 0.021

9 0.012 19 0.035

10 0.05 20 0.028

Table 6. The probability of a 50-week scenario (π_(s )) for each week

weeks π_(s ) weeks π_(s ) weeks π_(s ) weeks π_(s ) weeks π_(s ) 

1 0.004 11 0.004 21 0.009 31 0.005 41 0.014

2 0.02 12 0.012 22 0.011 32 0.012 42 0.003

3 0.011 13 0.05 23 0.007 33 0.007 43 0.007

4 0.012 14 0.013 24 0.008 34 0.018 44 0.008

5 0.008 15 0.004 25 0.015 35 0.008 45 0.008

6 0.006 16 0.016 26 0.014 36 0.009 46 0.009

7 0.012 17 0.005 27 0.011 37 0.014 47 0.012

8 0.017 18 0.01 28 0.013 38 0.016 48 0.011

9 0.006 19 0.013 29 0.018 39 0.014 49 0.014

10 0.011 20 0.014 30 0.012 40 0.011 50 0.013

Table 7. Probability of 100-week scenario (π_(s )) for each week

weeks π_(s ) weeks π_(s ) weeks π_(s ) weeks π_(s ) weeks π_(s )

1 0.004 21 0.003 41 0.002 61 0.002 81 0.005

2 0.008 22 0.007 42 0.006 62 0.004 82 0.003

3 0.009 23 0.003 43 0.005 63 0.007 83 0.002

4 0.007 24 0.003 44 0.002 64 0.007 84 0.006

5 0.004 25 0.007 45 0.004 65 0.005 85 0.004

6 0.002 26 0.009 46 0.004 66 0.002 86 0.002

7 0.005 27 0.004 47 0.009 67 0.005 87 0.005

8 0.011 28 0.007 48 0.008 68 0.007 88 0.005

9 0.002 29 0.008 49 0.008 69 0.006 89 0.005

10 0.007 30 0.004 50 0.008 70 0.004 90 0.008

11 0.01 31 0.003 51 0.007 71 0.002 91 0.008

12 0.004 32 0.006 52 0.008 72 0.006 92 0.006

13 0.01 33 0.003 53 0.007 73 0.005 93 0.006

14 0.005 34 0.009 54 0.002 74 0.007 94 0.008

15 0.009 35 0.004 55 0.005 75 0.005 95 0.005

16 0.007 36 0.004 56 0.006 76 0.008 96 0.008

17 0.003 37 0.007 57 0.01 77 0.004 97 0.005

18 0.002 38 0.008 58 0.006 78 0.003 98 0.007

19 0.009 39 0.008 59 0.003 79 0.008 99 0.003

20 0.005 40 0.005 60 0.007 80 0.003 100 0.006

Figure 2. Three Efficient Frontiers for Scenario in MAD

Table 8. Details of the obtained solutions for 3 risk measures in 3 scenarios

Volatility risk measures S REGT Return S.P Optimal selection

SV 1 0.343 58.33 4 (x_85=0.042,x_113=0.355,x_121=0434,x_149=0.169)

 2 0.297 64.92 4 (x_15=0.061,x_63=0.236,x_121=0.53,x_146=0.173)

 3 0.145 71.18 5 (x_85=0.011,x_113=0.047,x_124=0.002,x_125=0.771,x_149=0.169)

MAD 1 0.299 42.12 6 (x_3=0.192,x_7=0.115,x_31=0.33,x_33=0.086,x_98=0.141,x_143=0.136)

 2 0.166 99.87 6 (x_14=0.121,x_18=0.086,x_31=0.433,x_100=0.089,x_122=0.161,x_142=0.11)

 3 0.104 115.65 6 (x_7=0.081,x_14=0.186,x_31=0.502,x_100=0.116,x_110=0.037,x_121=0.078)

SAD 1 0.325 26.54 4 (x_64=0.408,x_100=0.098,x_101=0.165,x_149=0.329)

 2 0.311 40.09 6 (x_17=0.213,x_18=0.108,x_20=0.12,x_43=0.23,x_108=0.123,x_103=0.206 )

 3 0.137 84.15 6 (x_14=0.18,x_31=0.296,x_33=0.131,x_78=0.101,x_95=0.11,x_100=0.182)

S= Scenario, REGT= Regret total, S.P= Selected Portfolio in Optimize mode

To:

Reviewer 1- Comments 7

And

Reviewer 3- Comments 3

And

Reviewer 2- Comments 4

Reference Data type Model type Example Type Constraints Period Number Solution Technique Year Authors 

 Uncertainly Certainly Hypothetical Case Study Numerical Others Turnover Transaction Boundary Cardinality Multi-period Single period Simulation Metaheuristic algorithm Heuristic algorithm Exact solution 

 Others Stochastic Fuzzy Robust 

[3]

 � NLP � � � � bandit 2023 Kagrecha et al 1

[4]

 � � � � � 2022 Ding and Uryasev 2

[5]

 � MOLP � � � � � 2022 Groetzner and Werner 3

[6]

 � LP � � � � � � � � 2022 Benati and Conde 4

[7]

 � LP � � � � GA 2022 Caçador et al. 5

[8]

 � � LP and NLP � � � MC 2022 Filho and Silva Neiro 6

[9]

 � LP � � � � � 2021 Li et al. 7

[10]

 � LP � � � � � 2021 Gong et al. 8

[11]

 � LP � � � � 2021 Chakrabarti 9

[12]

 � LP � � � � � � � 2021 Caçador et al. 10

[13]

 � LP � � � � � � 2020 Won and Kim 11

[14]

 � LP � � � � � 2020 Li and Wang 12

[15]

 � LP and NLP � � � � 2020 Hernandez and al Janabi 13

[16]

 � LP � � � � � � 2020 Caçador et al 14

[17]

 � LP � � � � 2019 Vohra and Fabozzi 15

[18]

 � LP � � � � 2019 Baule et al. 16

[19]

 � LP � � � � � 2018 Huang et al. 17

[20]

 � MILP � � � � 2018 Van den Broeke et al. 18

[21]

 � LP � � � � 2018 Simões et al. 19

[22]

 � MOLP � � � � � 2018 Rivaz and Yaghoobi 20

[23]

 � MINLP � � � � � 2017 Xidonas, et al.(b) 21

[24]

 � MILP � � � � 2017 Xidonas , et al.(a) 22

[25]

 � MILP � � � � � 2017 Mohr and Dochow 23

[26]

 � LP � � � 2017 Grechuk and Zabarankin 24

[27]

 � � NLP � � � GA 2013 Fernandez et al. 25

[28]

 � MILP � � � 2012 Lourenço et al. 26

[29]

 � MILP � � � � � � 2012 Bean and Singer 27

[30]

 � MILP � � � � � 2011 Gregory et al. 28

[31]

 � LP � � � � 2006 Giove et al. 29

[32]

 � NLP � � � � 2006 Nwogugu 30

 � MILP � � � � SBA � 2024 Larni- Fooeik et al 31

Genetic Algorithm (GA), Mont Carlo (MC), Linear Programming (LP), Mult objective Linear Programming (MOLP), Non-Linear Programming (NLP), Mix Integer Linear Programming (MILP), Mix Integer Non-Linear Programming (MINLP), Scenario Based Approach (SBA).

---

## [Decision Letter · Decision Letter 1]

15 Feb 2024

Stochastic Portfolio Optimization: A Regret-Based Approach on Volatility Risk Measures: An Empirical Evidence from The New York Stock Market

PONE-D-23-39475R1

Dear Dr. Emran Mohammadi,

We’re pleased to inform you that your manuscript has been judged scientifically suitable for publication and will be formally accepted for publication once it meets all outstanding technical requirements.

Kind regards,

Shazia Rehman, Ph.D.

Academic Editor

PLOS ONE

Additional Editor Comments (optional):

Reviewers' comments:

Reviewer's Responses to Questions

**Comments to the Author**

1. If the authors have adequately addressed your comments raised in a previous round of review and you feel that this manuscript is now acceptable for publication, you may indicate that here to bypass the “Comments to the Author” section, enter your conflict of interest statement in the “Confidential to Editor” section, and submit your "Accept" recommendation.

Reviewer #2: All comments have been addressed

Reviewer #3: All comments have been addressed

2. Is the manuscript technically sound, and do the data support the conclusions?

Reviewer #2: Yes

Reviewer #3: Yes

3. Has the statistical analysis been performed appropriately and rigorously? 

Reviewer #2: Yes

Reviewer #3: Yes

4. Have the authors made all data underlying the findings in their manuscript fully available?

Reviewer #2: Yes

Reviewer #3: Yes

5. Is the manuscript presented in an intelligible fashion and written in standard English?

Reviewer #2: Yes

Reviewer #3: Yes

6. Review Comments to the Author

Reviewer #2: I appreciate your efforts to incorporate all my comments. I recommend this manuscript for publication in Plos One journal.

Reviewer #3: The authors addressed carefully all comments and suggestions that I have recommended. The Manuscript looks now more clear and it ras significantly improved.

7. PLOS authors have the option to publish the peer review history of their article (what does this mean?). If published, this will include your full peer review and any attached files.

Reviewer #2: **Yes: **Prof. Dr. Dilawar Khan

Reviewer #3: No

---

## [Editor Report · Acceptance letter]

24 Feb 2024

PONE-D-23-39475R1 

PLOS ONE

Dear Dr. Mohammadi, 

I'm pleased to inform you that your manuscript has been deemed suitable for publication in PLOS ONE. Congratulations! Your manuscript is now being handed over to our production team.

Kind regards, 

on behalf of

Dr. Shazia Rehman 

Academic Editor

PLOS ONE